# ScaleMoE: Mixture-of-Experts for Scalable Continuous Control in Actor-Critic Reinforcement Learning

**Yi Ma** [1]  **Chenjun Xiao** [2]  **Hongyao Tang** [3]  **Yaodong Yang** [4]  **Jinyi Liu** [3]  **Jing Liang** [3]  **Jiye Liang** [1]

## Abstract

Scaling network remains a bottleneck in deep reinforcement learning (RL): simply enlarging actor–critic networks destabilizes training and soon saturates performance. Although recent monolithic architectures such as SimBa and BRC have shown that carefully designed inductive biases can enable positive scaling up to a certain size, their improvements plateau soon as model parameters grow further. This work introduces Scale-MoE, a scalable RL architecture that integrates Mixture-of-Experts (MoE) modules into both the actor and critic of modern continuous control algorithms. Two complementary gating schemes are studied: output-level aggregation of per-expert policies and Q-functions, and feature-level fusion of expert representations before a shared head. We instantiate ScaleMoE on two representative monolithic RL baselines: the single-task method SimBa and the multi-task method BRC. Experiments across the DeepMind Control Suite, Meta-World, and HumanoidBench show that progressively increasing the number of experts (up to 64) yields substantial improvements in returns, significantly outperforming monolithic networks of comparable or even greater parameter counts. Results demonstrate that ScaleMoE provides an efficient and effective scaling axis for deep RL in continuous control.

[1]School of Computer and Information Technology, Shanxi University [2]School of Data Science, The Chinese University of Hongkong, Shenzhen [3]School of Computer Software, Tianjin University [4] Department of Computer Science and Engineering, The Chinese University of Hong Kong. Correspondence to: Chenjun Xiao <chenjunx@cuhk.edu.cn>, Hongyao Tang <tanghongyao@tju.edu.cn>.

*Proceedings of the 43$^{rd}$ International Conference on Machine Learning*, Seoul, South Korea. PMLR 306, 2026. Copyright 2026 by the author(s).

## 1. Introduction

Deep reinforcement learning (RL) has yet to realize the scaling gains that have propelled advances in computer vision and NLP. Increasing model size in RL often *hurts* performance rather than improving it, as naive width or parameter scaling of actor–critic networks can induce overfitting and instability, obscuring clear scaling laws. For example, (Andrychowicz et al., 2021; Bjorck et al., 2021) report that standard agents plateau or degrade as parameters increase. To counteract this, researchers have introduced architectures that embed inductive biases or regularization. SimBa (Lee et al., 2024) shows that architectural modifications, e.g., observation normalization and residual connections, can yield *positive* scaling on continuous control benchmarks. In the multi-task regime, BRC (Nauman et al., 2025) leverages a high-capacity distributional critic and regularization to achieve state-of-the-art results across many tasks. While these studies demonstrate that scaling is possible with careful design, their *monolithic* nature, i.e., activating all parameters for every input, imposes a fundamental limit, causing performance to plateau as model size increases beyond a certain point.

In contrast, Mixture-of-Experts (MoE) enables conditional computation: a gating function activates only a subset of expert networks per input, decoupling total capacity from per-sample compute (Ye & Xu, 2023). MoE has been highly effective for scaling in NLP and vision by expanding parameters without proportional increases in computation (Zhou et al., 2022; Riquelme et al., 2021). Applying MoE to RL is therefore a natural step toward scalable performance. Recent studies demonstrate promising results in discrete-action settings: Obando-Ceron et al. incorporate soft MoE layers into value-based agents and observe improved scaling on Atari, while Willi et al. show that experts can specialize in multi-task RL to boost sample efficiency. In continuous control, several works explore MoE from different perspectives, including interpretable expert gating (Akrour et al., 2021), task-oriented perturbations for visual RL (Huang et al., 2025), and decision transformer scaling (Kong et al., 2025). However, these approaches primarily focus on policy specialization without joint actor-critic scaling. The most relevant prior work (Hendawy et al., 2023) employs orthog-

onal constraints to enhance actor and critic expert diversity but does not investigate scaling effects. Crucially, existing MoE-in-RL efforts have not fully discovered the potentials of scalable continuous control with joint MoE scaling and optimization of actor and critic networks.

We introduce **ScaleMoE**, a MoE architecture for *continuous-action* deep RL. ScaleMoE replaces a single actor and critic with multiple expert actors and critics, and employs a learned gating mechanism that dynamically selects and combines expert outputs. We present two integration variants suited to actor–critic learning: (i) *output-level* gating, which fuses Gaussian policy parameters (means and variances) and aggregates $Q$-values across the top-$K$ experts; and (ii) *feature-level* gating, which mixes penultimate-layer features with a shared output head. We instantiate ScaleMoE on a single-task Simba (Lee et al., 2024) agent and on a multi-task BRC (Nauman et al., 2025) agent, keeping the underlying learning algorithms unchanged, highlighting the method's generality.

Across DeepMind Control Suite hard tasks (dog and humanoid domains) and the HumanoidBench of high-dimensional continuous control, ScaleMoE delivers **significant scaling of return** with increasing model size than strong monolithic baselines. For example, scaling to eight actor and critic experts yields an average **60%** improvement over the single-expert baseline on *humanoid-run*, whereas simply scaling a single network with an equivalent parameter budget provides limited gains or even degrades performance. We also study computational trade-offs: while increasing the number of experts raises training time and memory, MoE with smaller experts can outperform a single large network at far lower resource cost. Notably, an eight-expert or sixteen-expert BRC (each critic expert width 512) surpasses a single-network BRC with critic width 4096 in overall return while using roughly one-eighth/one-fourth of the parameters. We further analyze plasticity and specialization: ScaleMoE exhibits a substantially lower dormant-neuron ratio (Sokar et al., 2023), indicating more effective capacity utilization. Meanwhile, the gating policy induces clear expert specialization by task context in the multi-task setting. These results suggest that conditional computation via MoE offers a practical path to unlocking parameter scaling in RL continuous control.

Our contributions are threefold:

- **ScaleMoE for continuous control.** To the best of our knowledge, ScaleMoE is the first to investigate the scaling phenomenon in continuous control through the dual application of Mixture-of-Experts modules to both the actor and the critic, enabling a systematic study of how model capacity influences performance.

- **Actor–critic-tailored integrations with broad appli-**

**cability.** We develop output-level gating and feature-level gating, and instantiate them as drop-in modules for SimBa (single-task) and BRC (multi-task) without modifying their learning algorithms.

- **Validated scaling, plasticity, and efficiency.** On DMC hard tasks, MetaWorld and HumanoidBench, ScaleMoE exhibits superior *scaling of return*, lower dormant-neuron ratios with clear expert specialization, and favorable compute/memory trade-offs.

## 2. Related Work

### 2.1. Scaling Up Deep RL Architectures

Architectural innovations have enabled more stable training of wider networks in deep RL. Residual connections and normalization layers mitigate gradient pathologies in value functions, e.g., spectral normalization in SpectralNet (Bjorck et al., 2021) and batch renormalization in CrossQ (Bhatt et al., 2024)—while simplicity-biased designs such as SimBa (Lee et al., 2024) and SimBaV2 (Lee et al., 2025) reduce overfitting in overparameterized policies. BRC (Nauman et al., 2024) pointed out that strong regularization paired with optimistic exploration could lead to effective scaling of the critic networks. Further, Nauman et al. proposed to apply BRO architecture into multi-task RL with the design of categorical loss of value networks. In contrast, systematic *depth* scaling remains under-explored for reward-guided training. Notable exceptions demonstrate feasibility in niche settings: DT-VINs (Wang et al., 2024) train planning networks with up to 5,000 layers via adaptive highway losses, and self-supervised RL (Wang et al., 2025) reports emergent locomotion with 1,024-layer networks. However, these monolithic architectures often struggle to further improve performance as model size increases beyond a certain point.

### 2.2. Mixture-of-Experts (MoE)

Mixture-of-Experts (MoE) scales parameters through *expert routing*, activating only a subset of experts per input to decouple model capacity from per-sample compute (Ye & Xu, 2023). For discrete control, MoE has outperformed dense layers on Atari (Obando-Ceron et al., 2024) and shows expert specialization and improved sample efficiency in multi-task settings (Willi et al., 2024). In continous control, Akrour et al. proposed a MoE framework for continuous actions using interpretable experts with probabilistic gating. Huang et al. introduce task-oriented perturbations for visual RL with MoE, focusing on representation learning in pixel-based domains. Kong et al. scale decision transformers via MoE for massive multi-task learning. While these demonstrate the versatility of MoE architectures, they prioritize only policy specialization without joint actor-critic scaling.

Most relevant to our method is (Hendawy et al., 2023), which focus on employ orthogonal constraints to enhance expert diversity, but scaling effect is not investigated in this work. Our work distinguishes itself by introducing MoE to *both* actor and critic networks to achieve *scalable* continuous control, with top-$K$ gating mechanisms compatible with Gaussian policies and value aggregation.

Ensemble methods offer an alternative to single-model scaling by aggregating diverse components. Q-ensembles address exploration (Osband et al., 2016) and estimation biases (Lan et al., 2020). To improve sample efficiency, REDQ (Chen et al., 2021), DroQ (Hiraoka et al., 2022) and AQE (Wu et al., 2022) inject uncertainty with limited additional compute. Policy ensembles also show promise: ACE (Zhang & Yao, 2019) combines actor outputs via tree search, POLTER (Schubert et al., 2023) distills historical policies for unsupervised RL, and SAPG (Singla et al., 2024) partitions rollouts across parallel policies and reweights data via importance sampling. Unlike these ensembles that activate all components, MoE provides conditional computation benefits by activating only sparse subsets, achieving ensemble-like diversity at lower computational cost.

# 3. ScaleMoE: Scalable Mixture-of-Experts into Actor-Critic

Our goal is to incorporate a Mixture-of-Experts module into an actor-critic agent, so that the agent's capacity (number of parameters) can be increased via multiple expert networks, while a gating mechanism decides which experts to use for each state. We first describe the generic MoE actor-critic architecture, then detail two variants of the gating mechanism, and finally outline specific architectural choices for single-task and multi-task implementations.

## 3.1. MoE Actor-Critic Architecture

**Multiple Actors and Critics:** Instead of a single policy network (actor) and single value network (critic), ScaleMoE maintains a set of $N$ actor networks $\{\pi_i\}_{i=1}^N$ and $N$ critic networks $\{Q_i\}_{i=1}^N$ (or value functions $V_i$, depending on the algorithm). Each actor $\pi_i$ outputs the actions of a stochastic policy or a deterministic policy. For stochastic policy, the output is typically the mean $\mu_i(s)$ and standard deviation $\sigma_i(s)$ of a Gaussian distribution for action $a$ given state $s$. In the following part, we will describe our method based on stochastic policy. Each critic $Q_i$ outputs an estimate of the return (e.g. $Q_i(s,a)$). All experts share the same network architecture design as the baseline, but potentially with **reduced per-expert width** so that total parameters remain reasonable. For example, if the baseline critic has width 1024, using $N = 4$ experts of width 512 each yields roughly comparable total parameters. In single-task ScaleMoE, we keep each expert's architecture identical to the baseline's,

whereas in multi-task ScaleMoE we often shrink individual expert size to allow a larger number of experts without blowing up computation (see Section 4).

**Gating Network:** A learnable gating network $G(s)$ takes the state or some representation of the state as input and produces a set of $N$ gating scores $\{g_1, \ldots, g_N\}$ for the experts. We implement $G$ as a small feedforward network (two layers) that outputs $N$ values, followed by a softmax to obtain gating weights:

$$w_i(s) \; = \; \frac{\exp(g_i(s))}{\sum_{j=1}^N \exp(g_j(s))}, \quad i = 1, \ldots, N. \quad (1)$$

Here $w_i(s) \in (0, 1)$ and $\sum_i w_i(s) = 1$. To encourage specialization and efficiency, we use a **top-$K$ gating** strategy: the gating network selects the $K$ highest-weight experts for the current state, and sets the other weights to zero. The chosen weights are then renormalized to sum to 1, denoted $\tilde{w}_i$ for selected experts. This means at most $K$ experts are active per state, reducing computation if $K \ll N$. Given the gating output, the actor-critic combine the expert networks' outputs. We consider two approaches, illustrated in Figure 1.

## 3.2. Output-Level Gating (Late Fusion)

In this approach, each expert actor $\pi_i$ independently outputs a distribution $\mathcal{N}(\mu_i(s), \Sigma_i(s))$, where $\Sigma_i$ is typically diagonal with elements $\sigma_{i,1}^2, \ldots, \sigma_{i,d}^2$ for $d$-dimensional action. Similarly each expert critic produces a scalar $Q_i(s,a)$. The gating weights then fuse these outputs into a single policy and single value:

**Mixture of policies:** We form the final policy as a Gaussian with mean and variance given by a weighted combination of the expert means and variances. Specifically, if $\mathcal{K}(s)$ is the set of top-$K$ experts for state $s$, we compute

$$\mu_{\text{final}}(s) \; = \; \sum_{i \in \mathcal{K}(s)} \tilde{w}_i(s)\, \mu_i(s), \quad (2)$$

$$\sigma_{\text{final}}^2(s) \; = \; \sum_{i \in \mathcal{K}(s)} \tilde{w}_i(s)\, \sigma_i^2(s), \quad (3)$$

for each component of the action vector. This yields a single Gaussian policy that approximately represents the mixture of expert policies. Intuitively, the gating network "blends" the mean actions of the selected expert policies, e.g., for $K = 1$, this simply picks the single best expert's policy.

While we aggregate means and variances independently, we note that the resulting mixture is not a true Gaussian mixture due to ignored covariances. However, in practice, the diagonal covariance assumption is common in policy parameterization, and our experiments show the approach remains highly effective. For deterministic policies, the mixture of policies are similar to that of Q values below.

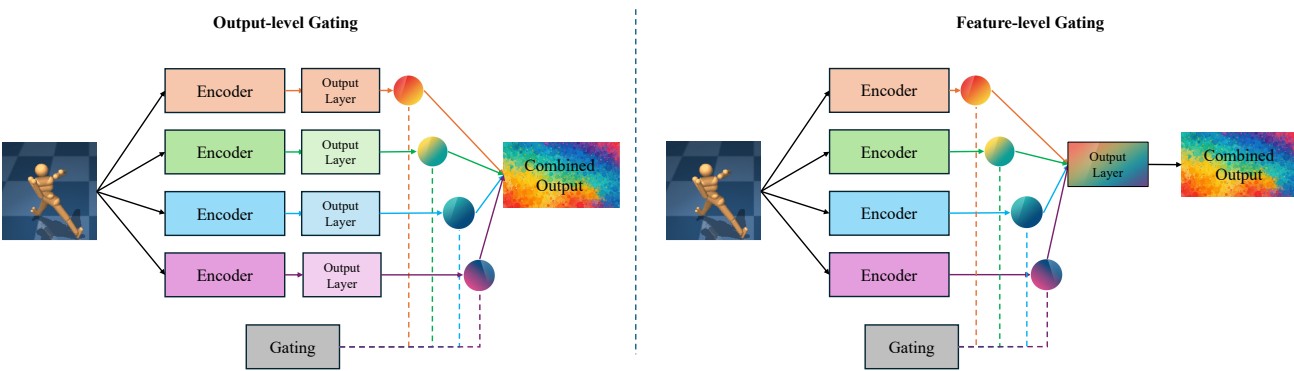

*Figure 1.* The illustration of two different level of gating in ScaleMoE. The architecture works both for actor and critic. The encoder part used in our experiments are Simba and BRC encoder.

**Mixture of $Q$-values:** For the critic, we similarly take the weighted sum of the selected experts' estimates:

$$Q_{\text{final}}(s, a) = \sum_{i \in \mathcal{K}(s)} \tilde{w}_i(s, a) \, Q_i(s, a). \qquad (4)$$

Note that the actor's gating is conditioned solely on the state, while the critic's gating is conditioned on the state-action pair, as the Q-function must route based on the specific action being evaluated to properly assess its value. The final $Q$ is a convex combination of the expert critics' outputs. Since each $Q_i$ is trained to estimate returns, the mixture $Q_{\text{final}}$ remains a valid estimate (and is used in the actor-critic loss as usual).

**Training.** All expert networks and the gating network are trained jointly. We apply standard RL losses $\mathcal{L}_{\text{RL}}$ (e.g., actor and critic losses) using the final fused $Q_{\text{final}}$ and the final policy. Note that gradients are only backpropagated through the top-K experts, while non-selected experts receive no gradient update for the current sample. This encourages specialization while avoiding over-training of rarely used experts.

To ensure that all experts are trained more evenly, we also add two *auxiliary regularizers*. Let $p_i^{(b)}$ denote the (post-softmax) gating probability of expert $i \in \{1, \dots, N\}$ for sample $b \in \{1, \dots, B\}$ in a minibatch, and define the batch-averaged usage $u_i = \frac{1}{B} \sum_{b=1}^{B} p_i^{(b)}$. We use a *load-balancing* loss (negative entropy of expert usage) to encourage uniform utilization across experts,

$$\mathcal{L}_{\text{lb}} = \sum_{i=1}^{N} u_i \log\big(u_i + 10^{-10}\big), \qquad (5)$$

and an *importance* loss to discourage overly peaked gating and reduce single-expert dominance,

$$\mathcal{L}_{\text{imp}} = \frac{1}{N} \sum_{i=1}^{N} \sum_{b=1}^{B} \big(p_i^{(b)}\big)^2. \qquad (6)$$

The total objective is

$$\mathcal{L}_{\text{total}} = \mathcal{L}_{\text{RL}} + \lambda_{\text{lb}} \, \mathcal{L}_{\text{lb}} + \lambda_{\text{imp}} \, \mathcal{L}_{\text{imp}}, \qquad (7)$$

where $\lambda_{\text{lb}}$ and $\lambda_{\text{imp}}$ control the strength of the regularizers. Together they ensure all experts receive sufficient gradient coverage, akin to balanced assignment in MoE literature (Zoph et al., 2022).

### 3.3. Feature-Level Gating (Early Fusion)

Our second integration method moves the gating operation to an earlier point in the network. Instead of merging final outputs, we merge the experts' *internal representations* before the output layer. Concretely, we let each expert produce a **penultimate-layer feature** $h_i(s)$ (e.g. the activations of the last hidden layer in the actor/critic network). The gating network $G(s)$ again selects top-$K$ experts and provides weights $\tilde{w}_i(s)$. We then compute a gated feature as the weighted sum of expert features:

$$h_{\text{gated}}(s) = \sum_{i \in \mathcal{K}(s)} \tilde{w}_i(s) \, h_i(s). \qquad (8)$$

This aggregated feature is then passed through a **shared output layer** that is used to produce the policy or value outputs. In practice, this means we have one additional fully-connected layer or small neural network that takes $h_{\text{gated}}(s)$ and outputs *both* $\{\mu_{\text{final}}(s), \sigma_{\text{final}}(s)\}$ for the actor and $Q_{\text{final}}(s, a)$ for the critic.

**Training:** Training is similar, except now the gating influences the experts' contributions at the feature level. The shared output layer's gradients flow back into the selected experts' features $h_i$ and into the gating network. This method ties the actor and critic experts more closely as they share the gated representation. In addition, this approach uses fewer total parameters than output-level gating, because the final layer is not duplicated across $N$ experts.

### 3.4. Single-Task vs Multi-Task Implementation Design

ScaleMoE is a algorithm-agnostic general module and we apply it to two different base algorithms SimBa and BRC.

**Single-Task (SimBa + MoE):** In this setting, we integrate MoE into a DDPG-style off-policy agent enhanced with the SimBa architecture. The baseline SimBa network has certain normalization and residual blocks, which we preserve in each expert. All hyperparameters (learning rates, layer sizes, etc.) are kept the same as in the original DDPG+SimBa setup, except for the addition of multiple networks and update-to-data ratio. We typically increase the number of actors and critics $N$ (e.g. from 1 to 4 or 8), while keeping each expert's size the same as the original network. This increases total parameters roughly linearly with $N$. For gating, we found a small network (two-layer MLP with 64 units) was sufficient to act as the gating network $G(s)$. The gating network in single-task case takes the state observation as input. It outputs $N$ gating logits as described earlier.

**Multi-Task (ScaleMoE BRC):** The multi-task setting uses the BRC agent as the backbone. BRC is a SAC-style actor-critic method that can be trained on many tasks simultaneously by using a large categorical value function conditioned on a task embedding. In our implementation, we extend BRC by introducing multiple critic experts and actor experts. We found it beneficial to **reduce the per-expert network width** compared to the original BRC architecture, in order to accommodate more experts without excessive increase in total parameters. For example, the original BRC critic has a layer width 4096 neurons; in ScaleMoE-BRC we use 8 critic experts each of width 512, which yields much fewer total neurons. For the actor network, we keep the width 256 unchanged. Despite each expert being smaller, the combined model has high capacity due to the number of experts. The gating network in multi-task takes as input both the state and the task identifier or task embedding. In practice, we concatenate the task embedding provided by BRC with the state features as input to the gating network. The rest of the training procedure follows BRC's training protocol.

**Algorithmic Pseudocode:** *Due to space limitation, we summarize at a high level:* At each training step, for a given state (and task, if multi-task), the gating network computes $w_i(s)$ and selects top-$K$ experts. The actor outputs (either each expert's policy or the shared output policy) produce the final action distribution, from which an action is sampled for environment interaction (for on-policy steps) or used in critic loss (for off-policy update). The critic outputs from experts are combined to get $Q_{\text{final}}$ which is used in the loss (e.g. temporal difference error). Gradients update the selected top-$K$ experts and gating network. Detailed algorithm is given in the Appendix A.

## 4. Experiments

### 4.1. Benchmarks and Tasks

We use the DeepMind Control Suite (DMC) Hard benchmark tasks (Tassa et al., 2018), including DMC DOGS (quadruped) and DMC HUMANOIDS (humanoid). For ScaleMoE with Simba, we train each task individually, while for BRC, we treat the entire DMC HUMANOIDS and DMC DOGS as a multi-task learning problem. Additionally, we test multi-task learning using HumanoidBench (Sferrazza et al., 2024), which includes 20 tasks for a simulated humanoid robot (Unitree H1), all trained concurrently with a task indicator. We also evaluate on MetaWorld (McLean et al., 2025a), a benchmark with 50 tasks for a robotic arm, treating them as a single multi-task problem.

### 4.2. Baselines

**Baselines:** For single-task experiments on DMC, our baseline is SimBa (DDPG) without MoE. Note that all the reported results of Simba is under the same update-to-data ratio with ScaleMoE. For multi-task DMC and Humanoid-Bench, the baseline is BRC, a strong multi-task agent that achieved state-of-the-art results on these benchmarks. We compare against BRC configured with its recommended network size (critic width 4096) and small sizes (critic width 512) to see if ScaleMoE is more parameter-efficient than simply using a bigger MLP. All the reported results are IQM and standard error of the proposed method and baselines averaged across eight seeds.

### 4.3. Results: Scaling Performance with Number of Experts

**Single-Task (Seven independent DMC Hard):** Figure 2 illustrates the scaling performance on the seven DMC hard tasks. With the baseline (SimBa) architecture, increasing the network size to critic hidden size of 512 had been shown to improve performance on these tasks (Lee et al., 2024), but naive scaling beyond that led to plateaus or degradation as indicated using the black dashed lines. In contrast, ScaleMoE shows a clear upward trend as we add more experts. For example, going from 1 expert (original Simba) to 4 experts (each same size as baseline) on ScaleMoE-Feature improves the average IQM by 32% on *dog-run*. On *humanoid-run*, using 8 experts yields gains up to 60%. Note that in each ScaleMoE method presented in Figure 2 uses a half active set out of all experts, which is sufficient to achieve overall scaling in performance. This suggests that the gating network is effectively routing each state to a good candidates of experts. We also note that both gating strategies (output-level and feature-level) perform similarly on single tasks, with a slight advantage for feature-level gating in final performance.

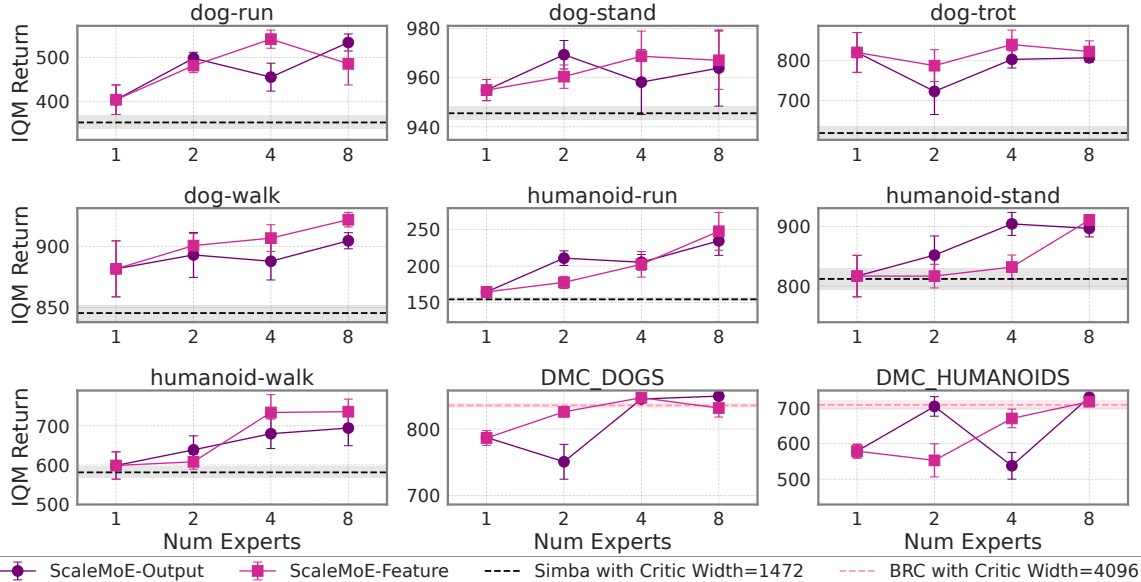

Figure 2. Scaling performance as the number of experts increases. Solid lines indicate results obtained by expanding from an identically sized monolithic architecture (critic hidden size of 512) to different number of experts. The black dashed line denotes the Simba with critic hidden size of 1472, which matches the model size of ScaleMoE with 8 experts of critic hidden size of 512. The pink dashed line denotes the original BRC configuration, i.e., a critic hidden size of 4096.

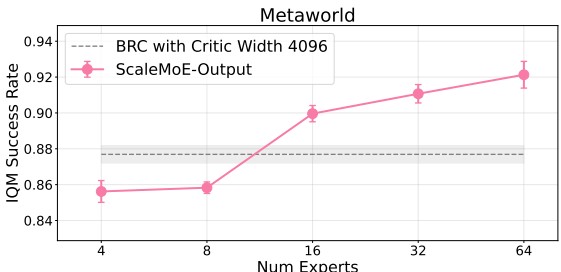

Figure 3. ScaleMoE scaling curves on 50-task MetaWorld.

**Multi-Task (DOGS 3 tasks and HUMANOIDS 4 tasks):**
The last two plots in Figure 2 also presents the average success rates across the three tasks in DMC DOGS and the four tasks in DMC HUMANOIDS, comparing our method with the baseline (BRC). The baseline BRC, employing a very large critic network (width 4096), achieves IQM of 835 on DOGS and 709 on HUMANOIDS, as indicated by the pink dashed lines. In contrast, our ScaleMoE-BRC model with 8 experts (each of width 512) attains final average IQM scores of 849 and 730, respectively. This results showed that ScaleMoE exceed the performance of the monolithic network. We also provide the model size comparison in Appendix F. ScaleMoE with 8 experts (actor + critic roughly 20M parameters) uses only 15% of the parameters of the default monolithic BRC (actor + critic roughly 136M), achieving a significant reduction in model size. This result is notable: instead of scaling a single network to 4096 units, employing *multiple smaller experts with a gating mechanism not only*

*reduces memory usage but also leads to better performance.* See appendix for the detailed total model parameter number of different architectures and expert numbers.

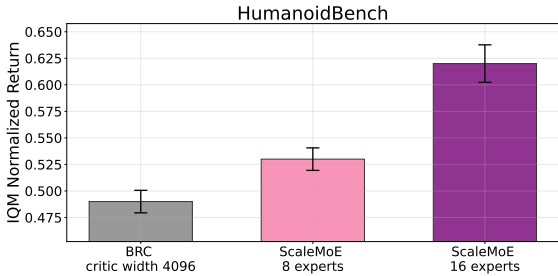

Figure 4. ScaleMoE results on 20-task HumanoidBench.

**Multi-Task (Metaworld (50 tasks) and HumanoidBench (20 tasks).** We conduct additional experiments on the MetaWorld benchmark, which involves 50 tasks. We activate only half of the total experts to scale the total expert number $N$. As shown in Figure 3, ScaleMoE with more than 16 experts (IQM $\geq 0.9$) significantly outperforms BRC-4096 (IQM = 0.88), demonstrating clear scaling benefits. In Figure 4, we also validate the effectiveness of scaling on HumanoidBench, where ScaleMoE with 16 experts of critic width 512 and full activation achieves an IQM normalized return of 0.62, significantly surpassing BRC (critic width 4096) at 0.49 (+27% relative improvement) with much fewer parameters. See Appendix K for training curves.

Building on prior work demonstrating the benefits of scal-

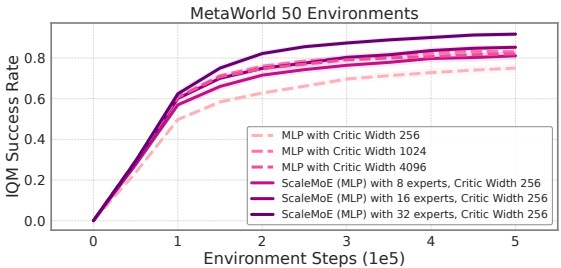

*Figure 5.* MLP-based ScaleMoE on 50-task MetaWorld.

ing simple feed-forward networks (McLean et al., 2025b) in multi-task RL, we replace the BRC backbone with an MLP for further experimental validation. Results in Figure 5 show that while scaling a monolithic network's width to 4096 yields diminishing returns, our ScaleMoE approach scaling the expert count to 32 achieves significantly better performance with notably fewer parameters, further justifying the efficiency and effectiveness of MoE-based scaling.

### 4.4. Analysis

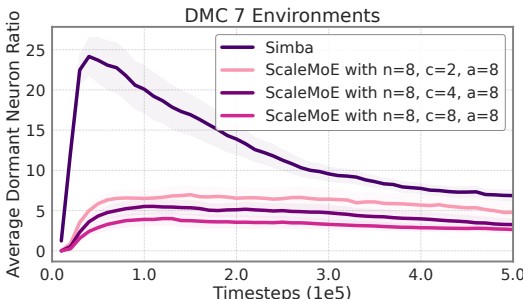

*Figure 6.* DNR during training.

**Network plasticity.** We then computed the dormant neuron ratio (DNR) for ScaleMoE (DDPG-Simba) with different activated number of critic expert during training in Figure 6. For the baseline single network Simba, we found a high dormant ratio, indicating a lot of capacity was unused, consistent with findings that networks in RL often don't utilize all units. In contrast, ScaleMoE with 8 experts had a significantly lower dormant ratio. Even with a conservative activation of $K = 2$ experts, the dormant ratio dropped to approximately 6% at step 5e5. A critical finding is that increasing activation does not lead to the loss of plasticity or stability typically associated with oversaturating a monolithic network. Combined results in Figure 9, we posit that the gating network essentially learns to decompose the input space, fostering **functional specialization** among experts. Each expert thereby specializes in distinct regions or modes of the data distribution. This structure enables coherent parameter updates and learns a more powerful internal representation, explaining the robust performance gains across

diverse domains. The lower dormant ratio thus signifies not merely more active parameters, but the emergence of an efficiently organized computational system. For other in-depth analysis, please refer to the appendix.

### 4.5. Scaling network width of monolithic baseline.

We provide monolithic baselines (BRC with increasing widths: 512/1024/2048/4096) alongside ScaleMoE variants (4/8/16/32/64 experts at fixed width 512) to enable direct comparison at matched parameter budgets in Figure 7. The table below reveals starkly different scaling behaviors: ScaleMoE exhibits consistent monotonic improvement, outperforming naive width scaling. Specifically, ScaleMoE with only 16 experts already surpasses the standard BRC baseline (width 4096). This comparison establishes that ScaleMoE's gains arise from efficient conditional computation via sparse routing rather than merely increasing parameter count through feedforward width expansion, validating that MoE architectures scale more effectively than monolithic widening.

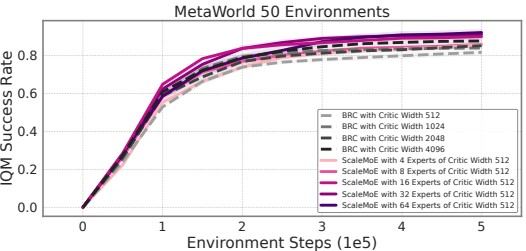

*Figure 7.* Comparison with monolithic baseline with different width.

### 4.6. Comparison with ensemble networks.

We implemented ensemble of BRC. As shown in Figure 8, ScaleMoE demonstrates favorable scaling properties that naive ensembling lacks. While Ensemble-BRC suffers from performance degradation as size increases, ScaleMoE achieves monotonic improvement.

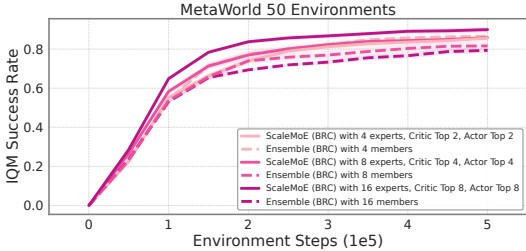

*Figure 8.* Comparison with ensemble of BRC.

**Expert utilization.** Figure 9 illustrates the **expert activation frequencies** observed during the evaluation of Scale-

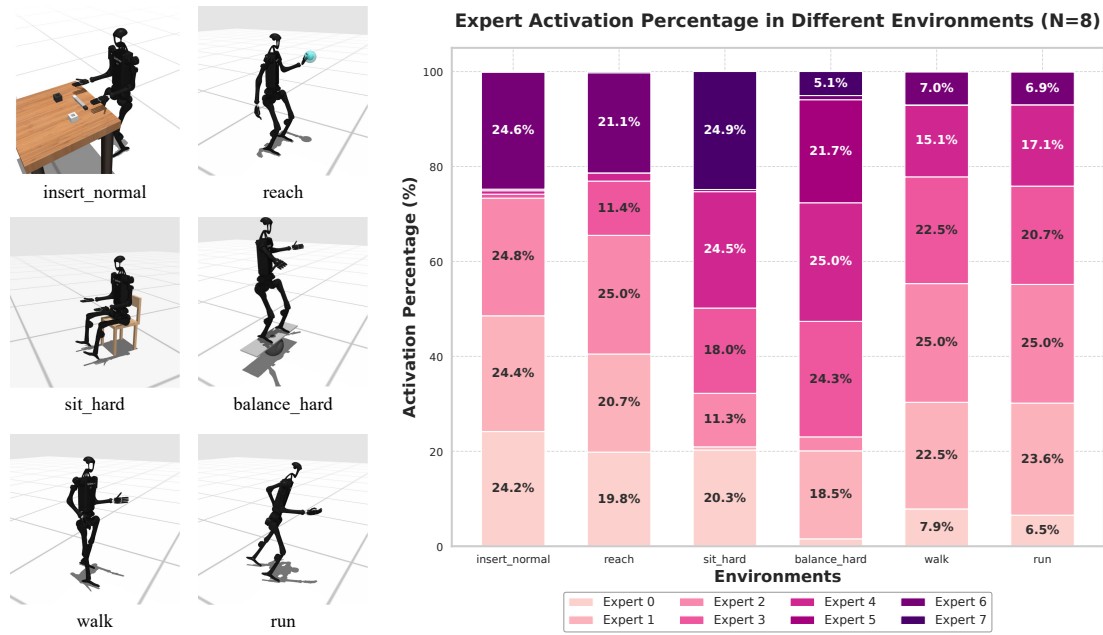

*Figure 9.* Expert activation heatmap on six representative tasks of HumanoidBench. Each cell shows the percentage of gating selections for an expert on a task. Percentage larger than 5% is marked.

MoE (with 8 experts, Critic Top 4, Actor Top 4) on representative tasks from HumanoidBench. The activation patterns are task-specific and reflect clear functional specialization: in locomotion tasks (walk, run), experts 1 to 3 (and occasionally expert 4) are predominantly selected, with expert 2 being the most frequently activated. For manipulation-oriented tasks (insert, reach), experts 0, 1, 2, and 6 are emphasized. Balance and sit tasks exhibit distinct routing behaviors: balance task primarily engages experts 1, 3, 4, 5, while sit task relies mainly on experts 0, 3, 4 and 7. Across tasks, the activation mass is typically distributed among 3 to 4 experts, each accounting for approximately 20–25% of the selections. This suggests that the gating mechanism effectively blends complementary skills rather than collapsing to a single dominant expert. Such task-adaptive co-activation underscores a key advantage of the MoE architecture: the gating network decomposes the multi-task problem into distinct subskills (e.g., locomotion, manipulation, and balance) and dynamically recombines them through top-K mixing.

## 5. Ablation Study

**Gating Position (Output vs Feature-Level):** We compare the two integration strategies on the DMC (Figure 2). On single-task DMC, feature-level (early-fusion) gating yields overall higher returns than output-level (late-fusion) gating. We conjecture that the shared output head in early fusion acts as a useful inductive bias: through a shared representation, it reduces variance in actor's Gaussian parameterization and critic's value estimation. In multi-task

experiments, however, the two strategies perform comparably. A hypothesis is that the expressivity of late fusion are largely offset by the parameter efficiency of early fusion. Thus, we favor feature-level gating for single task, while for multi-task settings we prefer the more easy-to-implement output-level gating variant.

**Number of Active Experts ($K$):** Figure 10 summarizes a sweep over $K \in \{1, 2, 4, 8\}$ with $N = 8$ experts of ScaleMoE-Output on both single-task and multi-task DMC. The overall best setting is fully expert activation. A lower-compute choice Top-2 and Top-4 on both actor and critic already attains satisfying performance. In contrast, when $K = 1$, many experts receive little to no gradient, leading to under-training and weak performance. These patterns suggest two practical insights. First, avoid $K = 1$, i.e., using at least two active experts per side prevents expert under-training and stabilizes learning. Second, scale actor and critic $K$ in tandem could bring more scaling benefits.

One may wonder if fully activating all experts achieves best performance, why do we still employ MoE? Three reasons: First, with sparse activation, the update frequency per expert decreases, potentially leading to undertraining; we address this by aligning the total update counts across different activation settings (Appendix I.1). Second, even under full activation, the gating mechanism functions as a SoftMoE, assigning differentiated weights rather than simply averaging all experts. Third, activating only a subset of experts enables significant computational savings in practice in Py-Torch implementations (Appendix G).

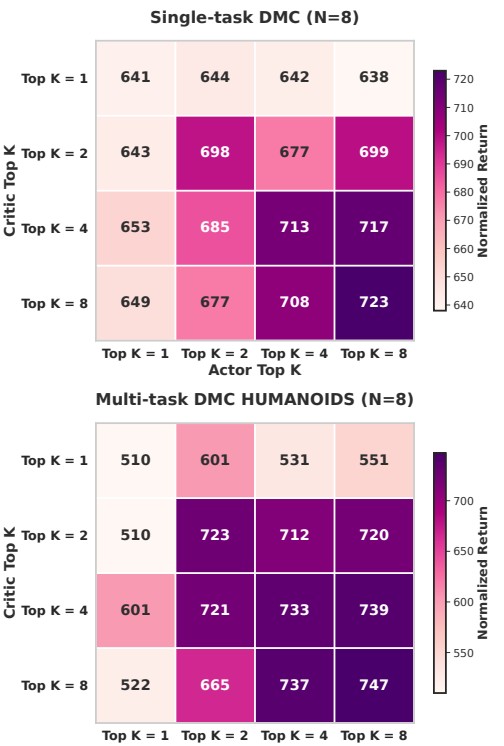

*Figure 10.* Normalized returns across different Top-K configurations on single tasks and multi tasks.

**Scaling Number of Experts ($N$):** We also ablated $N$ (total experts) while fixing $K = 2$ on DMC using ScaleMoE (Simba). Increasing the number of experts from $N = 2 \to 4 \to 8$ yields mean final scores of 675, 697, and 698, respectively. Thus, gains are monotonic but saturate around $N \approx 4$ under a fixed training budget. We hypothesize that with small $K$, additional experts are under-utilized: experience is partitioned more finely while only two experts receive gradients per state. In practice, when scaling $N$, one should also consider increasing $K$, extending training steps, or shrinking per-expert width to maintain sufficient data and gradient coverage per expert.

---

**Practical Guidance**

(1) Use feature-level gating for single-task and output-level for multi-task settings.
(2) Given $N$, scale $K \geq 2$ in tandem for both actor and critic for higher score.
(3) When scaling $N$, also adjust $K$, training steps, or network size to maintain expert utilization.

---

**MoE type ablation:** We implemented SoftMoE based on the BRC codebase. As shown in Figure 11 ScaleMoE consistently outperforms SoftMoE-BRC across all expert configurations. Currently the num_slots in SoftMoE is automatically determined as in (Obando-Ceron et al., 2024) and

could be very large under our large batch settings. Therefore, the unsatisfactory performance of it could derive from the unstable optimization. We will further investigate SoftMoE in continuous action spaces in our future work.

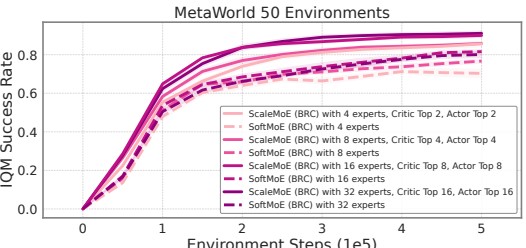

*Figure 11.* Ablation of MoE type.

**Auxiliary regularization ablation:** In our experiments, we use a single fixed value of $\lambda_{\text{lb}}$ and $\lambda_{\text{imp}}$, i.e., 0.01, across all tasks (MetaWorld 50 + Humanoid 20 + DMC 7) without any per-environment tuning. In practice, one could start with a minimal value and increase only as needed to ensure relatively balanced expert activation. The ablation in Figure 12 confirms stable performance across two orders of magnitude, indicating the parameter requires minimal tuning effort.

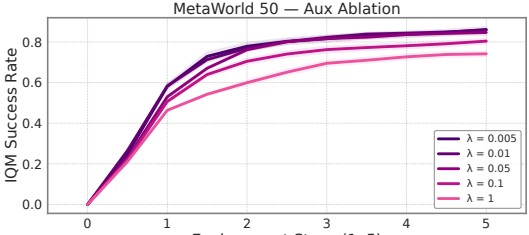

*Figure 12.* Ablation of auxiliary regularization weight.

## 6. Conclusion

We introduced ScaleMoE, a Mixture-of-Experts framework that brings conditional computation to continuous-control actor–critic RL. Increases model capacity in ScaleMoE enables scaling where monolithic networks plateau. The architecture is algorithm-agnostic: we demonstrated drop-in integrations with both single-task and multi-task baselines while preserving their original training procedures. Across diverse benchmarks, ScaleMoE delivered higher returns and revealed expert specialization. These findings establish MoE as a promising, compute-efficient pathway toward larger and more capable RL agents. We hope that ScaleMoE inspires further research on bridging the gap between the scale of models in supervised learning and those in reinforcement learning, ultimately enabling agents that learn **better, faster, and across many domains**.

## Acknowledgement

This work is supported by Fundamental Research Program of Shanxi Province (Serial No. 202503021212091), the National Natural Science Foundation of China (No.62406271, 62541610, U21A20473), the MoE Key Laboratory of Brain-inspired Intelligent Perception and Cognition, University of Science and Technology of China (No. 2521006).

## Impact Statement

This paper presents work whose goal is to advance the field of Machine Learning. There are many potential societal consequences of our work, none which we feel must be specifically highlighted here.

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

# A. Algorithm

---

**Algorithm 1** ScaleMoE: Mixture-of-Experts Actor-Critic Training

---

**Require:** Number of experts $N$, top-$K$ gating, replay buffer $\mathcal{D}$,
    learning rates $\alpha_{\text{actor}}, \alpha_{\text{critic}}, \alpha_{\text{gate}}$
1: Initialize actor experts $\{\pi_i\}_{i=1}^{N}$, critic experts $\{Q_i\}_{i=1}^{N}$, gating network $G$
2: **for** each training iteration **do**
3:     Sample batch $\{(s_j, a_j, r_j, s'_j, \tau_j)\}_{j=1}^{B}$ from $\mathcal{D}$    # $\tau_j$ only in multi-task
4:     **for** each sample $j = 1 \ldots B$ **do**
5:         Build gating input: $x_j = s_j$ (single-task) or $x_j = [s_j; \tau_j]$ (multi-task)
6:         $g_j \leftarrow G(x_j); \quad w_j \leftarrow \text{softmax}(g_j)$
7:         $\mathcal{K}_j \leftarrow \text{top-}K(w_j); \quad$ renormalize $\tilde{w}_{j,i}$ for $i \in \mathcal{K}_j$
8:         **if** Output-Level Gating (Late Fusion) **then**
9:             **for** $i \in \mathcal{K}_j$ **do**
10:                $(\mu_{j,i}, \sigma_{j,i}) \leftarrow \pi_i(x_j); \quad Q_{j,i} \leftarrow Q_i(x_j, a_j)$
11:             **end for**
12:             $\mu_j \leftarrow \sum_{i \in \mathcal{K}_j} \tilde{w}_{j,i}\mu_{j,i}; \quad \sigma_j \leftarrow \sum_{i \in \mathcal{K}_j} \tilde{w}_{j,i}\sigma_{j,i}; \quad Q_j \leftarrow \sum_{i \in \mathcal{K}_j} \tilde{w}_{j,i}Q_{j,i}$
13:         **else if** Feature-Level Gating (Early Fusion) **then**
14:             **for** $i \in \mathcal{K}_j$ **do**
15:                $h_{j,i} \leftarrow \text{encoder}_i(x_j)$
16:             **end for**
17:             $h_j \leftarrow \sum_{i \in \mathcal{K}_j} \tilde{w}_{j,i}h_{j,i}$
18:             $(\mu_j, \sigma_j) \leftarrow \text{shared\_actor\_head}(h_j); \quad Q_j \leftarrow \text{shared\_critic\_head}(h_j, a_j)$
19:         **end if**
20:     **end for**
21:     Compute RL loss $\mathcal{L}_{\text{RL}}$ with $Q_j, \mu_j, \sigma_j$
22:     $u_i \leftarrow \frac{1}{B} \sum_{j=1}^{B} w_{j,i}$ for $i = 1 \ldots N$
23:     $\mathcal{L}_{\text{lb}} \leftarrow \sum_{i=1}^{N} u_i \log(u_i + \epsilon); \quad \mathcal{L}_{\text{imp}} \leftarrow \frac{1}{NB} \sum_{j=1}^{B} \sum_{i=1}^{N} w_{j,i}^2$
24:     $\mathcal{L}_{\text{total}} \leftarrow \mathcal{L}_{\text{RL}} + \lambda_{\text{lb}}\mathcal{L}_{\text{lb}} + \lambda_{\text{imp}}\mathcal{L}_{\text{imp}}$
25:     Update all experts and $G$ via $\nabla\mathcal{L}_{\text{total}}$
26:     Update target networks (if used)
27: **end for**

---

# B. Hyperparameters

**Backbone algorithms: SimBa and BRC.** Our work is built upon the JAX codes provided in official Simba (Lee et al., 2024) and BRC (Nauman et al., 2025) code repository. All experiments are run on NVIDIA GeForce RTX 4090 and H200 GPUs with INTEL(R) XEON(R) PLATINUM 8563C. For single task, we choose DDPG-Simba as our algorithm backbone. SimBa (Lee et al., 2024) is a single-task, off-policy actor–critic agent built on SAC that emphasizes a simplicity-biased architecture to stabilize training at larger widths. Concretely, SimBa employs observation normalization, residual MLP blocks, and lightweight normalization in the actor and critic to mitigate overfitting and gradient pathologies, enabling positive performance scaling on continuous-control benchmarks without altering the underlying SAC objective. Except for the MoE design and the updates per step, the other hyperparameters are kept the same with the original implementation in DDPG-Simba. The hyperparameters are shown in Table 1.

For mutliple tasks, we choose SAC-BRC as our algorithm backbone BRC (Nauman et al., 2025) targets the multi-task regime with a high-capacity, regularized categorical value function (distributional critic) conditioned on task information, paired with a compatible actor. Its design increases critic expressivity while using regularization to reduce interference across tasks, yielding state-of-the-art results over large task suites. Except for the MoE design and the Critic hidden dimension, the other hyperparameters are kept the same with the original implementation in SAC-BRC. The hyperparameters are shown in Table 2.

*Table 1.* ScaleMoE (DDPG-Simba) Hyperparameter Settings

| | Hyperparameter | Value |
|---|---|---|
| | Critic block type | SimBa Residual |
| | Critic num blocks | 2 |
| | Critic hidden dim | 512 |
| | Critic learning rate | 1e−4 |
| | Target critic momentum | 5e−3 |
| | Actor block type | SimBa Residual |
| | Actor num blocks | 1 |
| **Architecture** | Actor hidden dim | 128 |
| **(DDPG Simba)** | Actor learning rate | 1e−4 |
| | Exploration noise | $\mathcal{N}(0, 0.1^2)$ |
| | Batch size | 256 |
| | Optimizer | AdamW |
| | Optimizer momentum | (0.9, 0.999) |
| | Weight decay | 1e−2 |
| | Discount | Heuristic |
| | Updates per step | 4 |
| | Action repeat | 2 |
| | Clipped Double Q | False |
| | Number of experts | 1, 2, 4, 8 |
| | Number of activated critic experts | 1, 2, 4, 8 |
| **MoE** | Number of activated actor experts | 1, 2, 4, 8 |
| | weight of load-balancing loss ($\lambda_{\mathrm{lb}}$) | 0.01 |
| | weight of importance loss ($\lambda_{\mathrm{imp}}$) | 0.01 |

*Table 2.* ScaleMoE (SAC-BRC) Hyperparameter Settings

| | Hyperparameter | Value |
|---|---|---|
| | Critic block type | BroNet |
| | Critic depth | 2 |
| | Critic hidden dim | 512 |
| | Critic learning rate | 3e−4 |
| | Target critic momentum | 5e−3 |
| | Clipped Double Q | True |
| | Actor block type | BroNet |
| | Actor depth | 1 |
| | Actor hidden dim | 256 |
| | Actor learning rate | 3e−4 |
| **Architecture** | Temperature learning rate | 3e−4 |
| **(SAC BRC)** | $V_{min}$ | -10 |
| | $V_{max}$ | 10 |
| | num atoms | 101 |
| | Buffer size per task | 1e6 |
| | Polyak $\tau$ | 5e-3 |
| | Target update frequency | 1 |
| | Batch size | 1024 |
| | Optimizer | AdamW |
| | Weight decay | 1e−4 |
| | Target entropy | $|\mathcal{A}|/2$ |
| | Action repeat | 1 |
| | Discount | 0.99 |
| | Updates per step | 2 |
| | Number of experts | 1, 2, 4, 8 |
| | Number of activated critic experts | 1, 2, 4, 8 |
| **MoE** | Number of activated actor experts | 1, 2, 4, 8 |
| | weight of load-balancing loss ($\lambda_{\mathrm{lb}}$) | 0.01 |
| | weight of importance loss ($\lambda_{\mathrm{imp}}$) | 0.01 |

## C. Limitations

While ScaleMoE shows promising results, there are several limitations to note. **First**, the improved performance comes at the cost of added complexity. Tuning an MoE architecture involves extra hyperparameters (number of experts $N$, gating network shape, $K$ value, etc.), which could be daunting. **Second**, our experiments focused on continuous control tasks with state observations of modest dimensionality (mostly low-dimensional proprioceptive states, possibly some egocentric

target encodings). We did not evaluate ScaleMoE on tasks with high-dimensional image observations; integrating MoE with deep convolutional encoders (e.g., for pixel-based RL) might pose additional challenges or require a different gating design. **Finally**, from a computational standpoint, while we demonstrated efficiency gains relative to a single huge network, training an MoE agent still requires more hardware resources than a standard small network. In settings where compute is very limited or real-time inference is required on embedded systems, an MoE might be impractical. Future work may explore distilling the MoE policy back into a smaller network for deployment.

## D. Score normalization

We follow the method used in BRC (Nauman et al., 2025) to normalize the returns in the range of 0 to 1 in HumanoidBench using:

$$\text{Normalized Returns} = \frac{\text{Returns} - \text{Random Returns}}{\text{Success Returns} - \text{Random Returns}} \tag{9}$$

Detailed random returns and success returna are given in Table 3.

*Table 3.* Random and success scores for HumanoidBench tasks.

| Task | Random Score | Success Score |
|---|---|---|
| h1hand-balance_hard-v0 | 10.032 | 800.0 |
| h1hand-balance_simple-v0 | 10.170 | 800.0 |
| h1hand-bookshelf_hard-v0 | 14.848 | 2000.0 |
| h1hand-bookshelf_simple-v0 | 16.777 | 2000.0 |
| h1hand-crawl-v0 | 278.868 | 800.0 |
| h1hand-hurdle-v0 | 2.371 | 700.0 |
| h1hand-insert_normal-v0 | 1.673 | 350.0 |
| h1hand-insert_small-v0 | 1.653 | 350.0 |
| h1hand-maze-v0 | 106.233 | 1200.0 |
| h1hand-pole-v0 | 19.721 | 700.0 |
| h1hand-reach-v0 | -50.024 | 1200.0 |
| h1hand-run-v0 | 1.927 | 700.0 |
| h1hand-sit_hard-v0 | 2.477 | 750.0 |
| h1hand-sit_simple-v0 | 10.768 | 750.0 |
| h1hand-slide-v0 | 3.142 | 700.0 |
| h1hand-spoon-v0 | 4.661 | 650.0 |
| h1hand-stair-v0 | 3.161 | 700.0 |
| h1hand-stand-v0 | 11.973 | 800.0 |
| h1hand-walk-v0 | 2.505 | 700.0 |
| h1hand-window-v0 | 2.713 | 650.0 |

## E. Resource Consumption

*Table 4.* Resource Consumption of ScaleMoE on DMC tasks.

| Metric | DDPG-Simba (DMC) | | | | SAC-BRC (DMC HUMANOIDS) | | | |
|---|---|---|---|---|---|---|---|---|
| | Number of Experts | | | | Number of Experts | | | |
| | 1 | 2 | 4 | 8 | 1 | 2 | 4 | 8 |
| GPU Memory (MB) | 750 | 1000 | 1500 | 2500 | 1100 | 1500 | 2000 | 3000 |
| Training Time (Hours) | 3.2 | 5.3 | 7.2 | 11.4 | 9.3 | 10.5 | 11.8 | 13.7 |

Table 4 shows the resource consumption of the ScaleMoE model on the single-task and multi-task DMC benchmark as the number of experts increases. Both GPU memory usage and training time grow when scaling from 1 to 8 experts. However, it should be noted that this consumption could be effectively converted to performance improvement as reported before.

## F. Model Size Comparison

Here we present the model parameters of Simba, BRC and ScaleMoE with different expert numbers.

*Table 5.* ScaleMoE (Simba) model parameters (M indicates million).

| Method | Total Model Parameters (Critic + Actor) | Number of critic blocks | Critic Width | Expert Number |
|---|---|---|---|---|
| Simba (Standard) | $\approx 16.65$M | 2 | 512 | 1 |
| ScaleMoE (with 2 Simba Standard expert) | $\approx 33.24$M | 2 | 512 | 2 |
| ScaleMoE (with 4 Simba Standard expert) | $\approx 66.44$M | 2 | 512 | 4 |
| ScaleMoE (with 8 Simba Standard expert) | $\approx 132.83$M | 2 | 512 | 8 |
| Simba (Larger) | $\approx 133.12$M | 2 | 1472 | 1 |

*Table 6.* ScaleMoE (BRC) model parameters (M indicates million).

| Method | Total Model Parameters (Critic + Actor) | Number of Critic Blocks | Critic Width | Expert Number |
|---|---|---|---|---|
| BRC (Small) | $\approx 2.51$M | 2 | 512 | 1 |
| ScaleMoE (with 2 BRC Small expert) | $\approx 5.01$M | 2 | 512 | 2 |
| ScaleMoE (with 4 BRC Small expert) | $\approx 10$M | 2 | 512 | 4 |
| ScaleMoE (with 8 BRC Small expert) | $\approx 19.99$M | 2 | 512 | 8 |
| ScaleMoE (with 16 BRC Small expert) | $\approx 39.97$M | 2 | 512 | 16 |
| ScaleMoE (with 32 BRC Small expert) | $\approx 79.91$M | 2 | 512 | 32 |
| ScaleMoE (with 64 BRC Small expert) | $\approx 159.81$M | 2 | 512 | 64 |
| BRC (Standard) | $\approx 136.25$M | 2 | 4096 | 1 |

## G. Computation Benefit of Using Top-K

In theory, top-(K) routing reduces the per-step compute of the expert towers roughly in proportion to (K/N). For example, with (N=8) experts, selecting (K=2) should yield 25% of the expert FLOPs; similarly (K=4) should yield 50%, matching standard MoE analyses. This is true in dynamic-execution frameworks such as PyTorch, where only the selected experts are executed. As verified by our own profiling, selecting top-2 from 100 experts is 8.67× faster than perform computing on all 100 experts. However, our implementation uses JAX/XLA, which does not support runtime dynamic indexing under JIT compilation. As a result, selective Top-K execution must still evaluate all experts and apply gating afterward. Therefore, in JAX implementation, Top-K acts as a modeling/optimization hyperparameter, not a compute-saving mechanism. Considering readers may try to reproduce of work in PyTorch, we normally choose to activate half of the experts to seek for balance between efficiency and performance in our paper

Importantly, saving time compute is not the primary purpose of ScaleMoE in our work. As analyzed before, the main benefit of ScaleMoE in continuous-control RL is better expert specialization (task- or state-region-wise) and lower critic target variance as (K) increases. These effects collectively improve optimization stability and final performance, even when wall-clock compute remains similar of different K under our current parallel implementation.

## H. In-depth Analysis

To further understand the underlying reasons for ScaleMoE's gains, we add three new quantifiable metrics to explain the benefits of ScaleMoE.

**Gradient Interference Between Experts.** The interference metric quantifies the degree to which the gradients from different experts conflict with each other during training. Specifically, it is measured by the cosine similarity between the gradients of the experts, denoted as $g_i$ for expert $i$ and $g_j$ for expert $j$, as:

$$\cos(g_i, g_j) = \frac{\langle g_i, g_j \rangle}{|g_i||g_j|} \tag{10}$$

where $\langle g_i, g_j \rangle$ is the dot product of the gradients and $|g_i|$ is the L2 norm of the gradient vector. A low interference value corresponds to low conflict between the gradients of experts, meaning that they are learning complementary representations. As more experts are activated (increasing (K)), we observe a decrease in interference, bringing the metric closer to 0, indicating that experts are working in harmony rather than conflicting. This reduction in interference improves gradient flow, stabilizing the learning process and allowing the model to efficiently leverage the expertise of different specialists without overlap. As shown in Figure 13, increasing (K) leads to more balanced gradient updates, which enhances learning efficiency, aligning with the theoretical analysis in Appendix L (D).

**Expert Activation Entropy.** We also assess the entropy of expert activation across different tasks or states. The activation

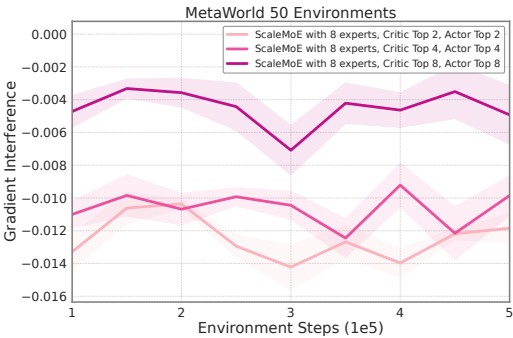

*Figure 13.* Gradient interference of ScaleMoE (BRC) on Metaworld 50 envs.

entropy $H$ is computed as:

$$H = -\sum_{i=1}^{N} p_i \log(p_i), \tag{11}$$

where $p_i$ is the probability of expert $i$ being activated, typically normalized by the gating weights. A high entropy value indicates that the gating mechanism distributes the activation more evenly across experts, ensuring that no single expert dominates the learning process, which would lead to under-utilization of the model's capacity. Figure 14 demonstrates that as the number of activated experts (K) increases, the entropy of expert activation increases, reflecting a more uniform expert usage. This balanced activation improves the model's ability to generalize and prevents any expert from being neglected, which is essential for multi-task learning and task specialization.

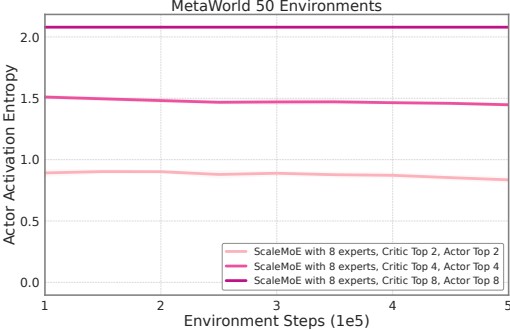

*Figure 14.* Actor expert activation entropy of ScaleMoE (BRC) on Metaworld 50 envs.

**Critic Target Variance.** The variance of the critic's target values is an important measure of the stability of the value function. We define the variance of the critic's target $y_K$ as:

$$\text{Var}(y_K) = \mathbb{E}[(y_K - \mathbb{E}[y_K])^2], \tag{12}$$

where $y_K$ is the target value predicted by the critic, which depends on the output of the top-K experts. As the number of active experts (K) increases, the variance of the critic's target decreases. This reduction in variance indicates that the critic is more stable and provides more reliable target estimates, which helps to improve convergence and learning efficiency. Lower variance in the target signal is beneficial for both actor and critic updates, leading to faster and more stable training. Empirical evidence supporting this can be seen in Figure 15, where we show that variance decreases as (K) increases, corroborating our theoretical findings in Appendix L (B).

## I. Additional ablation Studies

### I.1. Updating experts with the same number of gradient flows

Under our design with the load-balancing and importance loss, we ensure that experts are activated in a fairly balanced manner, which mitigates the risk of under-training. However, when using very small values of (K) (i.e., selecting only a

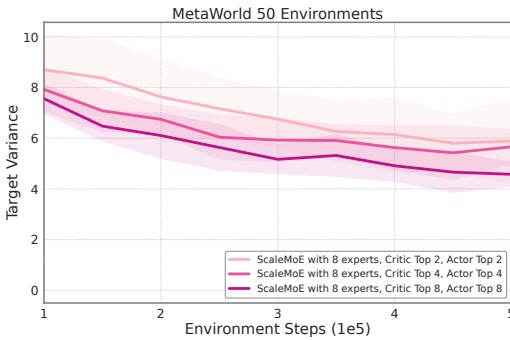

*Figure 15.* Critic target variance of ScaleMoE (BRC) on Metaworld 50 envs.

few experts per state), each expert is updated fewer times, leading to a situation where some experts may not be effectively trained. This is particularly problematic in the case of sparse activation.

To address this, we maintain a fixed total number of updates for all experts by increasing the update-to-data ratio when k is small, ensuring that each expert is updated equally. This prevents experts from becoming under-trained. Our experiments demonstrate that with each expert receiving the same updates, ultimately the agent with small k could also lead to similar performance with that with large k.

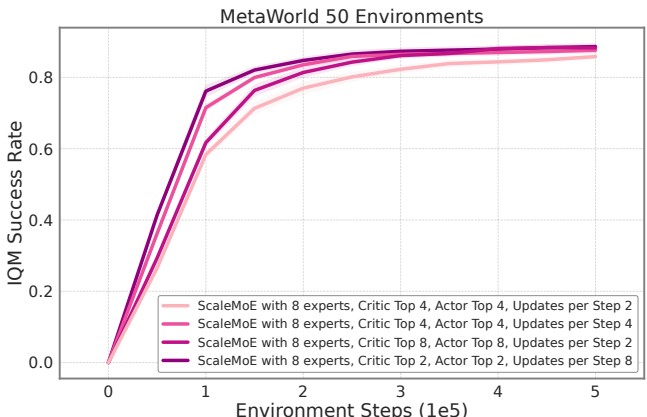

*Figure 16.* Ablation on experts updating times

It should be noted that even in the fully activated setting (Top-8/Top-8), the gating mechanism still plays a key role. The gating network does not simply activate all experts with equal weight, but instead, it assigns different weights to each expert based on the gating network's output. This means that, while all experts are being used, they are not used equally. Some experts may receive a larger weight in certain contexts, and others may be down-weighted accordingly.

### I.2. Actor expressivity ablation

From an intuitive perspective, in Figure 10, the improvement of increasing activated actor comes from two obvious factors (1) more thorough training: activating more experts ensures each expert gets more updates, preventing under-training. (2) the expressibility of the combined policy: more experts lead to a more expressive policy that can capture multi-modal behaviors, improving performance.

To prove the improved expressivity of the gated actor, we keep the activated critic expert number to 4 unchanged, and change the number of actor experts with full activation to ensure each expert is trained with same gradient steps. We show the results on Metaworld with 50 tasks in the Figure 17. It shows that as the actor expert number increases, the overall performance also increases. which is consistent with our theoretical analysis in Appendix L (E). To further explore the underlying reasons, we investigate whether increasing actor could bring other benefits. We investigate the Q-target variance in the critic. We found that as more experts are activated, the critic's target Q-values become more stable and reliable, leading to better convergence.

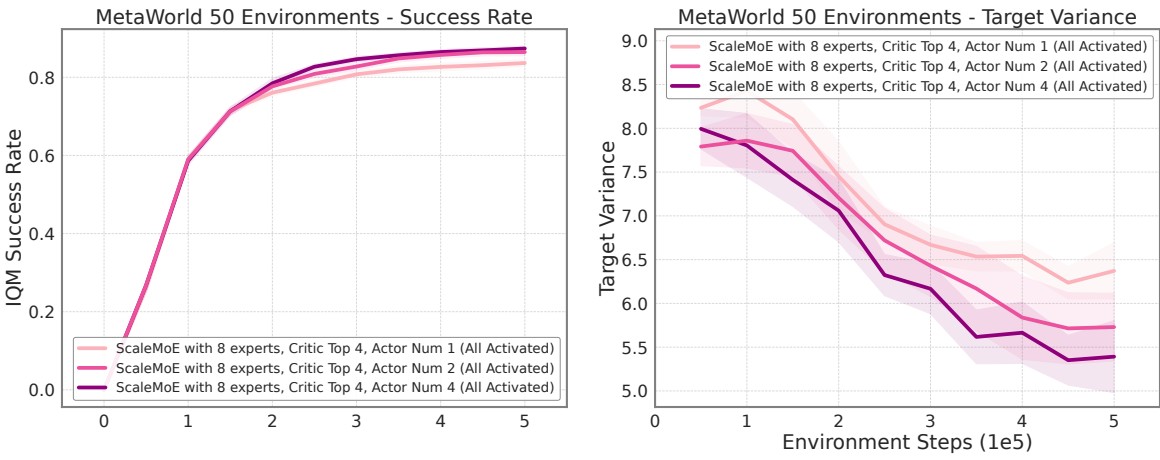

*Figure 17.* Ablation on actor expressivity.

## I.3. Auxiliary loss ablation

We compare the performance of ScaleMoE with and without the load-balancing and importance regularization loss. As expected, we found that with the auxiliary loss, the model performs better overall, especially in terms of stability and convergence in Figure 18.

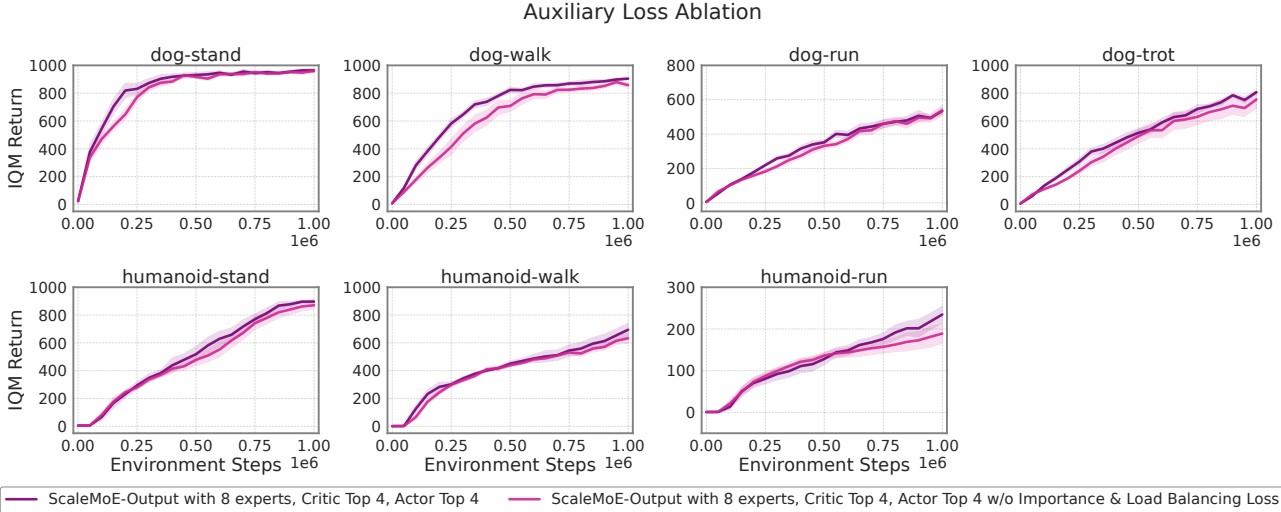

*Figure 18.* Ablation on load-balancing and importance loss.

To further validate the effectiveness of the load-balancing loss, we present an additional figure (Figure 19) that shows the expert activation frequency during training. This plot illustrates how often each expert is activated across all tasks. When the load-balancing loss is applied, we observe that expert activation is much more uniform across all experts. In contrast, without the load-balancing loss, some experts are hardly activated at all, which can lead to under-utilization. As a result, the network's overall representational capacity is significantly compromised, as certain experts fail to specialize and learn meaningful features.

In Figure 20, we also provide a comparison of expert activation frequencies of a single run in one task, highlighting how the activation pattern changes with and without the load-balancing loss. With the load-balancing loss, experts are evenly activated, ensuring that each expert contributes to the learning process. Without the loss, certain experts receive significantly fewer updates, leading to poorer task-specific performance and less effective task specialization.

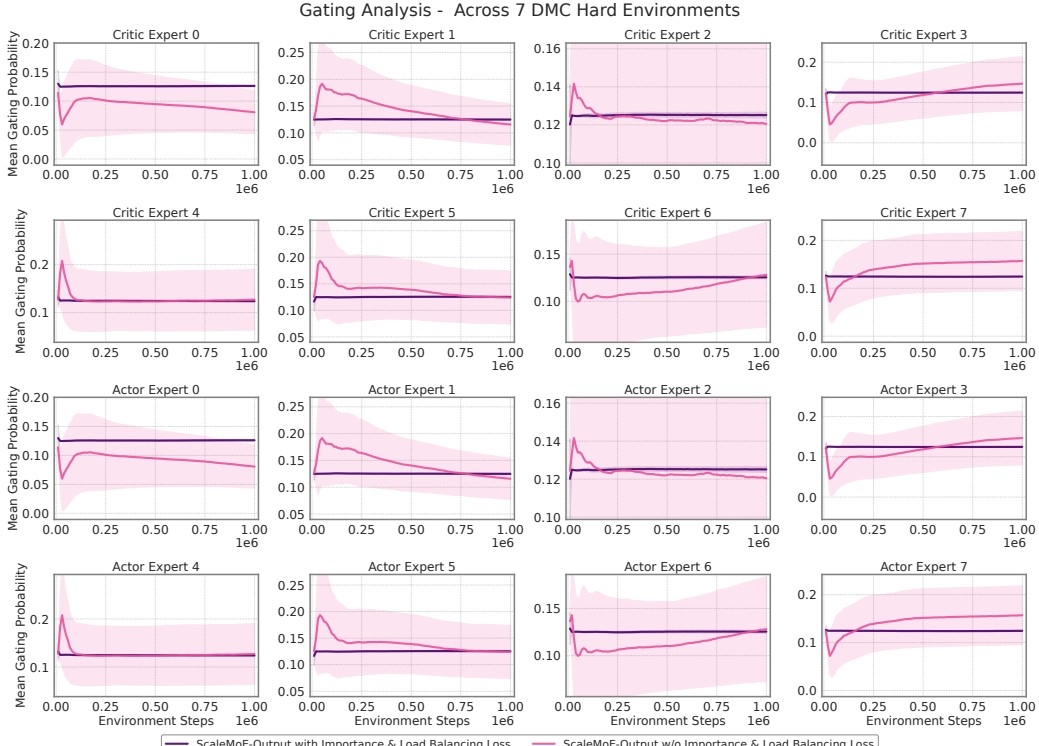

*Figure 19.* Expert activation during training on DMC Hard envs.

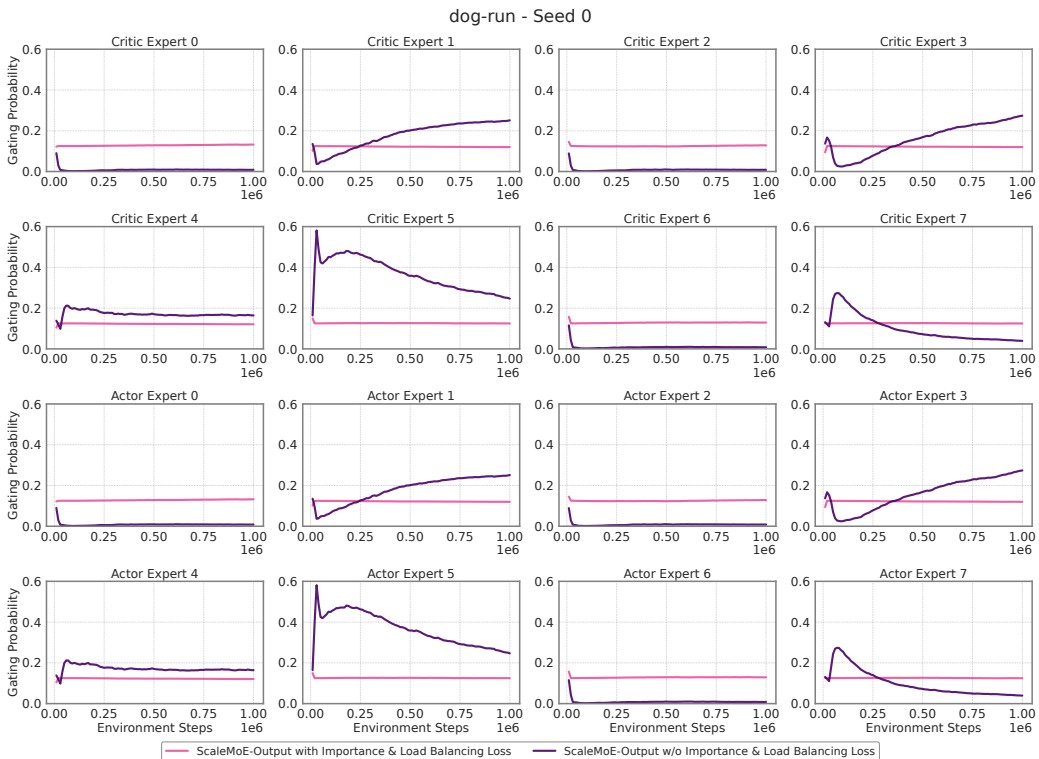

*Figure 20.* Expert activation during training on dog-run env.

# J. Median, IQM and Mean Results

We present median, IQM and mean results with 95% confidence interval in this section using the tool provided in this work (Agarwal et al., 2021).

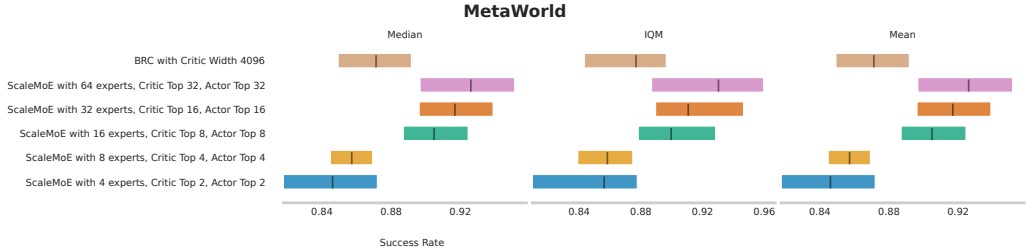

*Figure 21.* Median, IQM and Mean with 95% Confidence Interval on Metaworld.

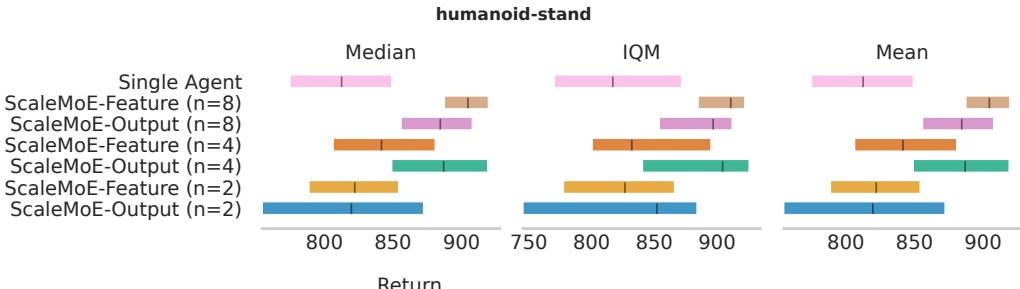

*Figure 22.* Median, IQM and Mean with 95% Confidence Interval on humanoid-stand.

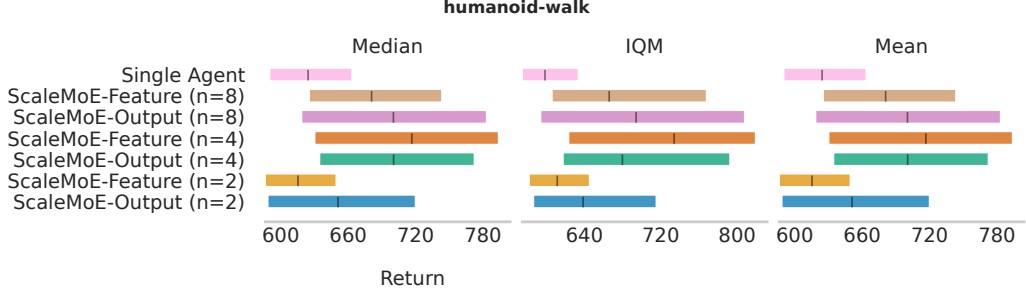

*Figure 23.* Median, IQM and Mean with 95% Confidence Interval on humanoid-walk.

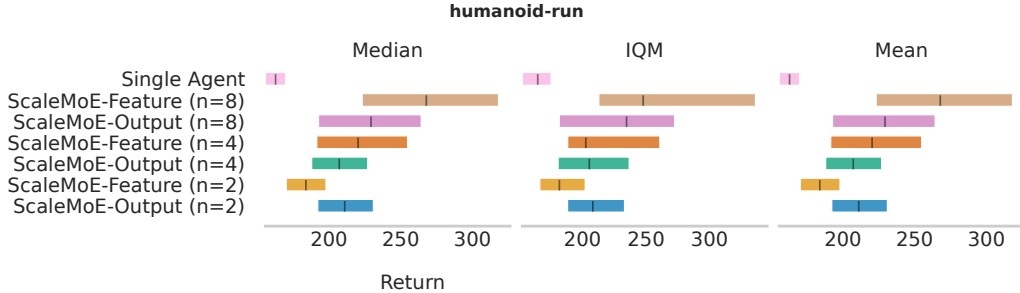

*Figure 24.* Median, IQM and Mean with 95% Confidence Interval on humanoid-run.

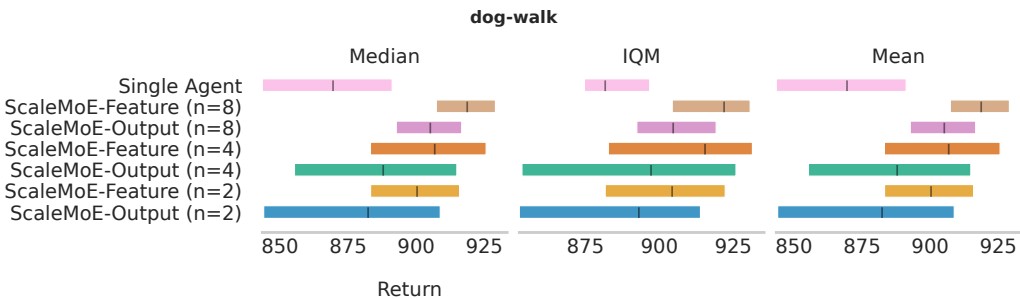

*Figure 25.* Median, IQM and Mean with 95% Confidence Interval on dog-walk.

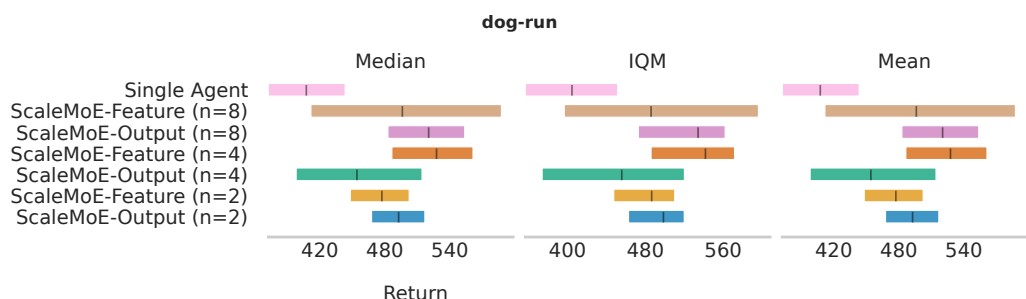

*Figure 26.* Median, IQM and Mean with 95% Confidence Interval on dog-stand.

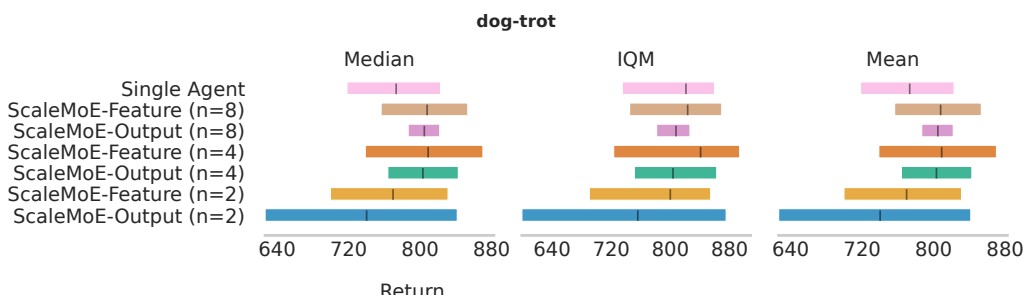

*Figure 27.* Median, IQM and Mean with 95% Confidence Interval on dog-trot.

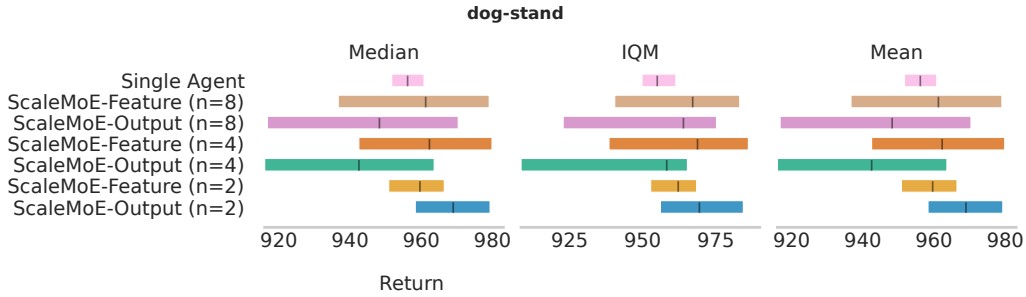

*Figure 28.* Median, IQM and Mean with 95% Confidence Interval on dog-stand.

# K. Training Curves

Here we provide the training curves of ScaleMoE and baselines.

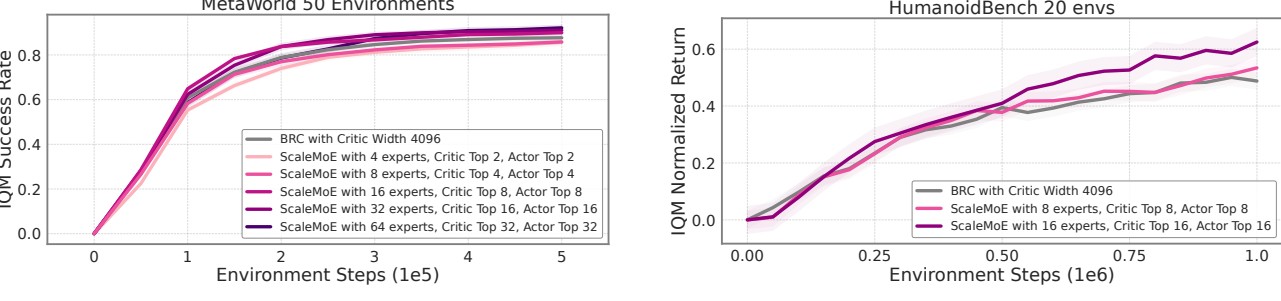

*(a)* Training curves on Metaworld 50 envs.  *(b)* Training curves on HumanoidBench 20 envs.

*Figure 29.* Scaling performance on multi-task environments.

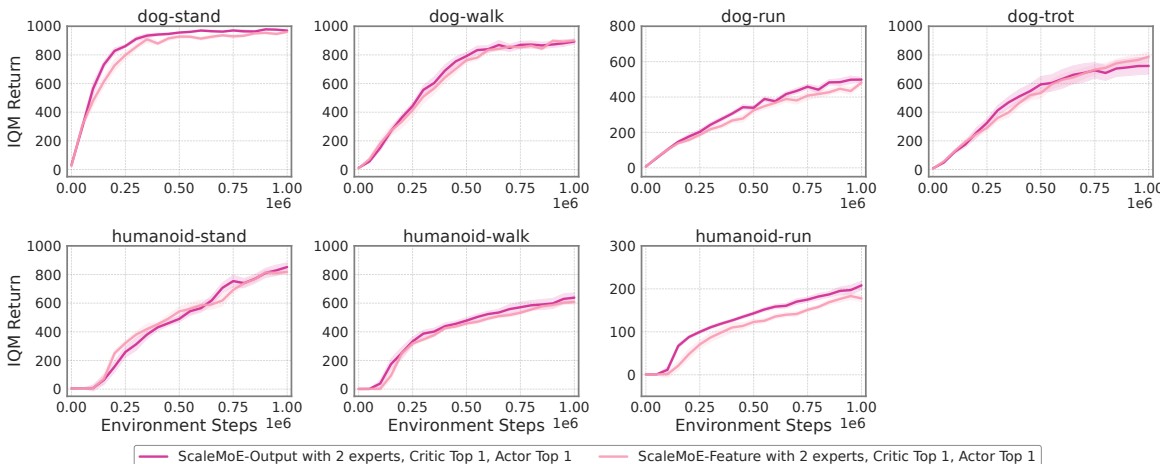

*Figure 30.* Single-task training curves of ScaleMoE (Simba) on each DMC env with two experts.

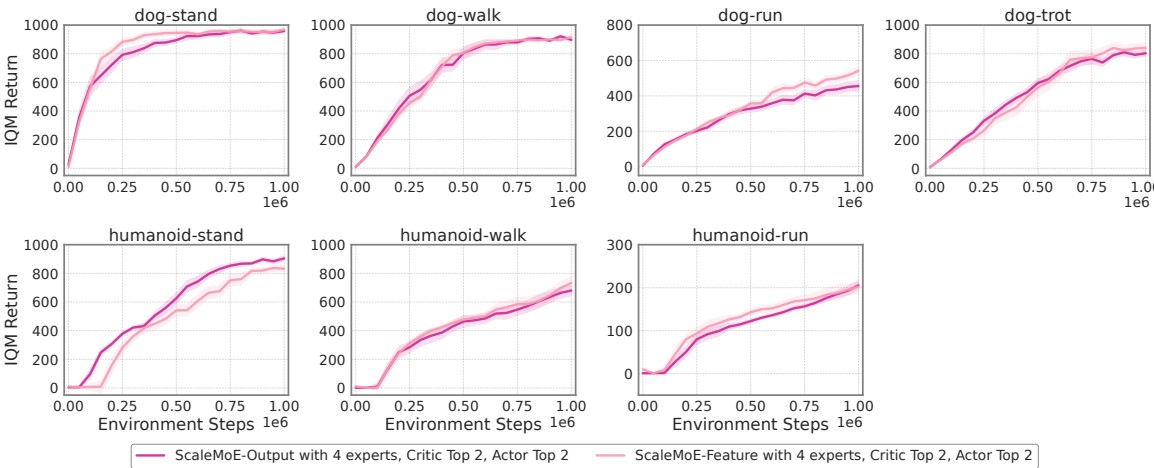

*Figure 31.* Single-task training of ScaleMoE (Simba) on each DMC env with four experts.

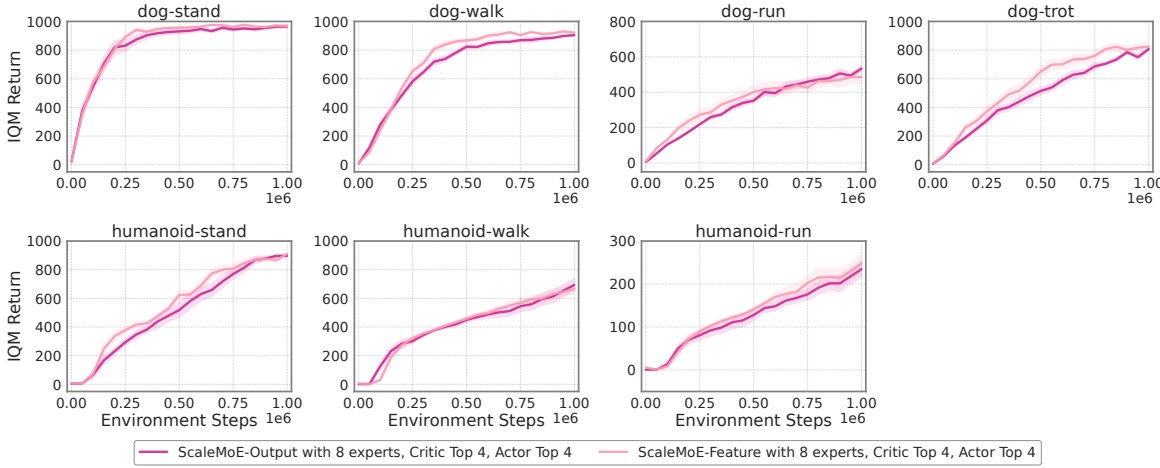

*Figure 32.* Single-task training curves of ScaleMoE (Simba) on each DMC env with eight experts.

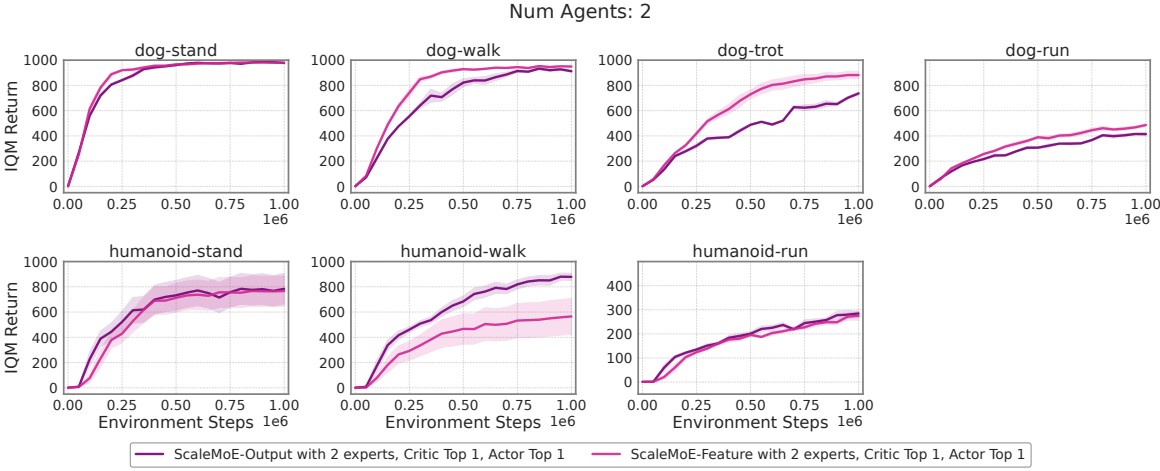

*Figure 33.* Multi-task training curves (normalized scores) of ScaleMoE (BRC) on each DMC env with two experts.

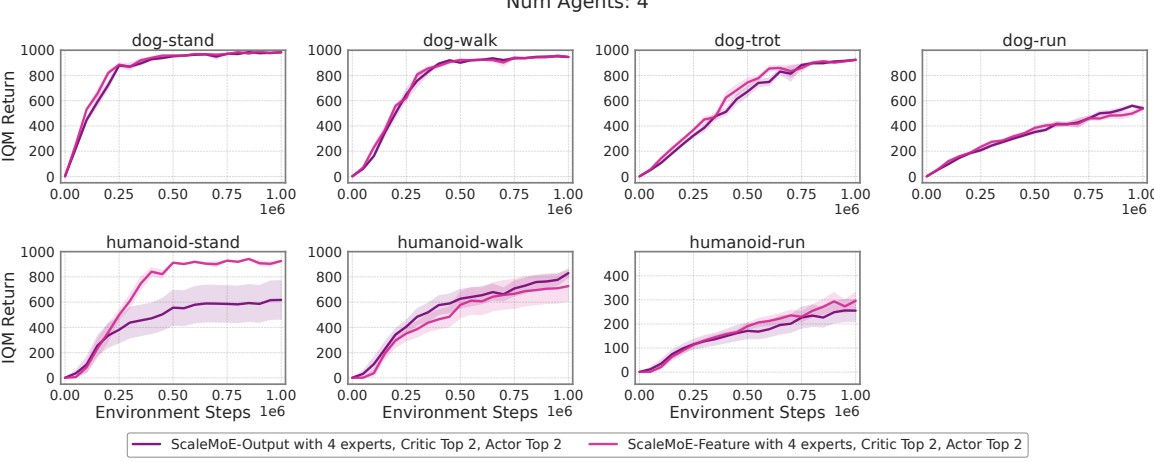

*Figure 34.* Multi-task training curves (normalized scores) of ScaleMoE (BRC) on each DMC env with four experts.

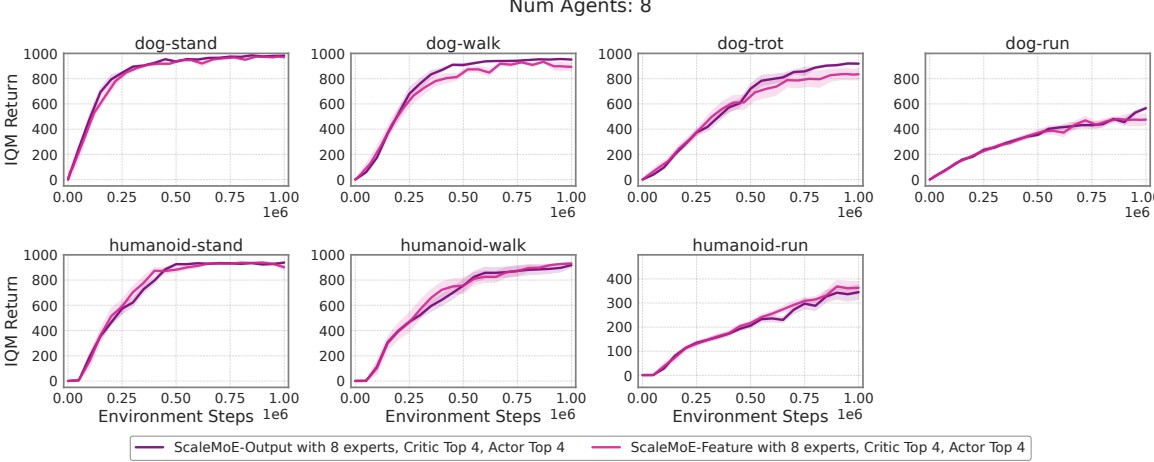

*Figure 35.* Multi-task training curves (normalized scores) of ScaleMoE (BRC) on each DMC env with eight experts.

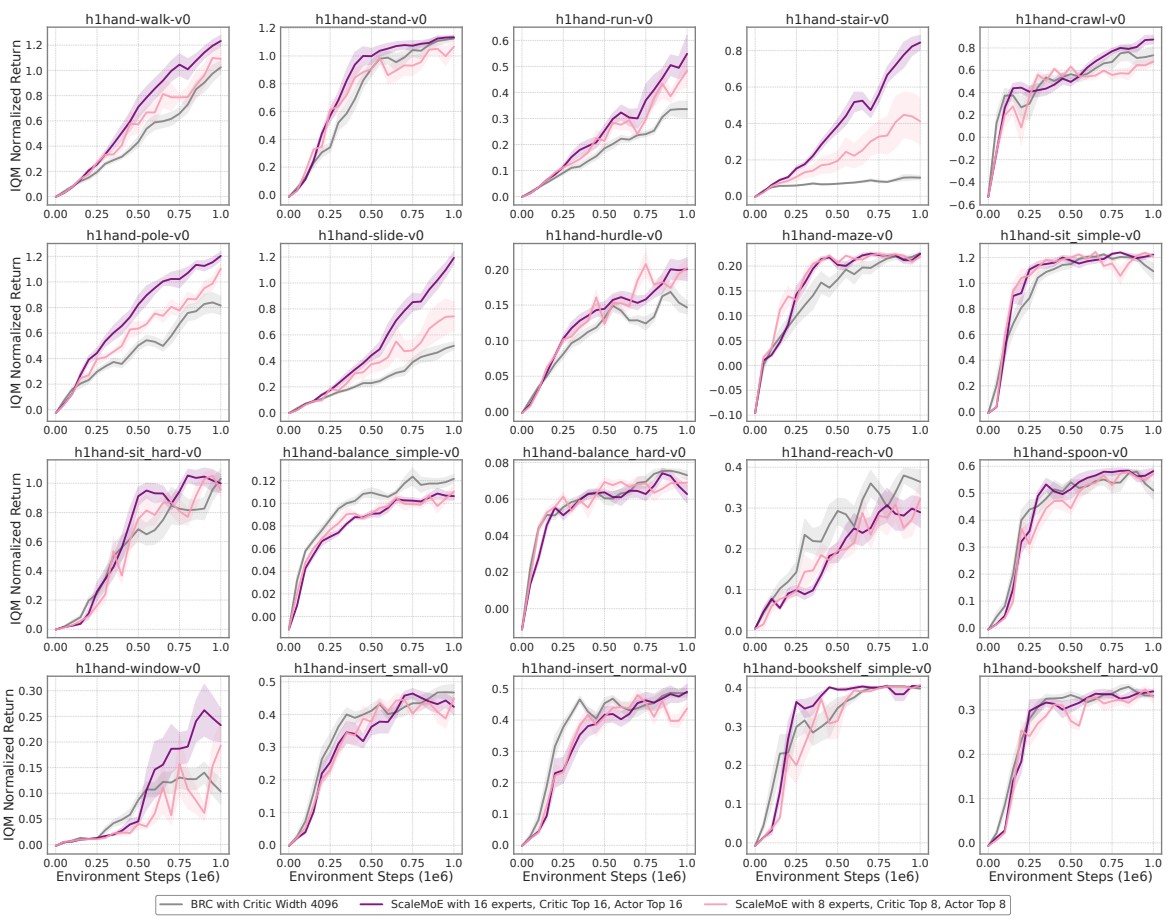

*Figure 36.* Multi-task training curves (normalized scores) of ScaleMoE (BRC) on each HumanoidBench env with eight experts.

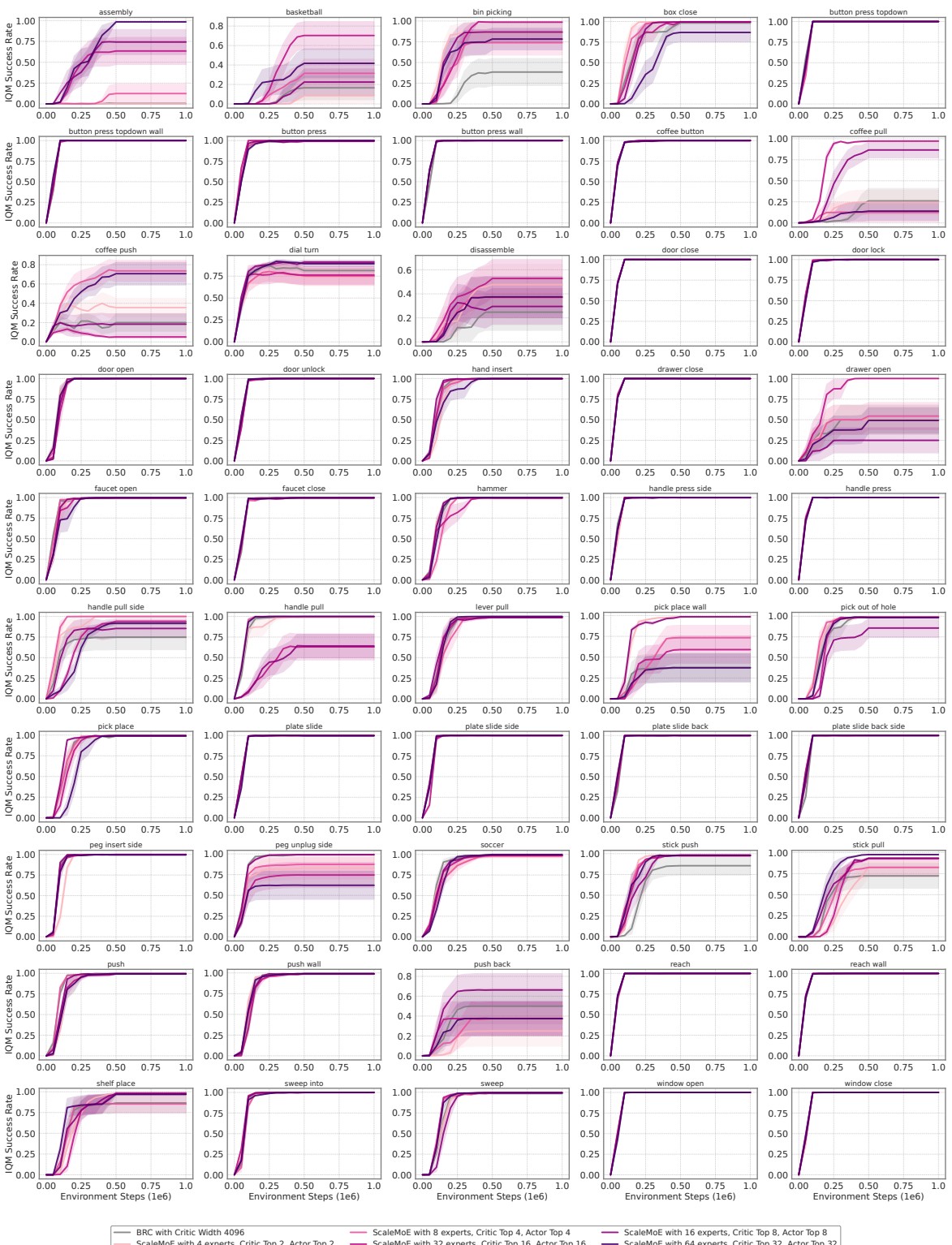

*Figure 37.* Multi-task training curves of ScaleMoE (BRC) on each MetaWorld env.

# L. Theoretical Intuition

We sketch several complementary arguments, i.e., approximation, variance, and optimization, that explain the gains we observe with ScaleMoE.

**(A) Specialization reduces approximation error.** Let $f^\star(s, a)$ denote the target function (a policy map or a $Q$-function). A soft-gated MoE induces a partition of unity $\{w_i(s)\}_{i=1}^N$ with $\sum_i w_i(s) = 1$, yielding

$$f_{\text{MoE}}(s, a) = \sum_{i=1}^N w_i(s) f_i(s, a).$$

If the state–action space decomposes into regions $\{\mathcal{R}_i\}$ on which $f^\star$ is simpler (e.g., lower curvature, lower intrinsic dimension), then fitting a smaller-capacity expert $f_i$ on $\mathcal{R}_i$ can reduce the local estimation error compared to a single monolithic model of the same total size. Formally, a local Rademacher complexity bound (Yin et al., 2019) gives a generalization gap of order

$$\mathbb{E}\big[\ell(f_{\text{MoE}}) - \ell(f^\star)\big] \lesssim \sum_{i=1}^N \underbrace{\mathfrak{R}_n(\mathcal{F}_i; \mathcal{R}_i)}_{\text{local complexity}} + \text{noise},$$

where $\mathfrak{R}_n(\mathcal{F}_i; \mathcal{R}_i)$ is smaller than the global complexity $\mathfrak{R}_n(\mathcal{F}_{\text{mono}})$ when $f^\star$ is piecewise-structured. Intuitively, the gate learns to route states to experts with the right inductive bias, yielding lower bias without inflating variance.

**(B) Variance reduction from top-$K$ mixtures.** For critic evaluation with squared loss, suppose each expert's TD target has zero-mean estimation noise $\varepsilon_i$ with $\text{Var}[\varepsilon_i] = \sigma_i^2$. Assume (i) the noises are pairwise uncorrelated, and (ii) the fusion weights $\tilde{w}_i(s)$ are non-negative and sum to one. Then the fused critic

$$Q_{\text{final}}(s, a) = \sum_{i \in \mathcal{K}(s)} \tilde{w}_i(s) Q_i(s, a)$$

satisfies

$$\text{Var}\big[Q_{\text{final}} \mid s, a\big] = \sum_{i \in \mathcal{K}(s)} \tilde{w}_i^2(s) \sigma_i^2 \le \Big( \max_{i \in \mathcal{K}(s)} \sigma_i^2 \Big) \sum_{i \in \mathcal{K}(s)} \tilde{w}_i^2(s).$$

Under equal weights ($\tilde{w}_i \equiv 1/K$) and identical variances ($\sigma_i^2 \equiv \sigma^2$) the bound becomes $\sigma^2/K$, showing a $K$-fold variance reduction relative to any single head. When correlations are present, the general expression

$$\text{Var}\big[Q_{\text{final}} \mid s, a\big] = \sum_{i \in \mathcal{K}(s)} \tilde{w}_i^2(s) \sigma_i^2 + 2 \sum_{i < j} \tilde{w}_i(s) \tilde{w}_j(s) \, \text{Cov}(\varepsilon_i, \varepsilon_j)$$

implies that positive correlations diminish the benefit, whereas negative correlations can further decrease the variance. [1] A similar argument applies to policy gradients: mixing multiple expert policies reduces gradient variance when experts are not positively correlated.

**(C) Contraction and stability under convex fusion.** For policy evaluation, the Bellman operator $T^\pi$ is a $\gamma$-contraction in $\|\cdot\|_\infty$ (Sutton & Barto, 2018). When the fusion weights depend on the *next* state, the update rule

$$Q_{\text{final}}(s, a) = \sum\nolimits_{i \in \mathcal{K}} w_i(s) Q_i(s, a)$$

must be analysed through the *joint* operator

$$(\mathcal{T}Q)(s, a) = r(s, a) + \gamma \sum_{s'} P(s'|s, a) \sum_{a'} \pi(a'|s') \sum_{i \in \mathcal{K}} w_i(s') Q(s', a').$$

Writing $\mathcal{T}$ as the linear map

$$(\mathcal{T}Q)(s, a) = r(s, a) + \gamma \sum_{s', a'} P_\pi(s', a'|s, a) \sum_{i \in \mathcal{K}} w_i(s') Q(s', a'),$$

---

[1] If the gates $\tilde{w}_i(s)$ are random (attention-based), the unconditional variance is $\mathbb{E}_s\big[\sum_i \tilde{w}_i^2(s)\sigma_i^2\big]$ under zero-covariance; online estimates can be obtained by sampling.

with $P_\pi(s', a'|s, a) = P(s'|s, a)\pi(a'|s')$, we have

$$\|\mathcal{T}Q - \mathcal{T}Q'\|_\infty \leq \gamma \max_{s,a} \sum_{s',a'} P_\pi(s', a'|s, a) \sum_{i \in \mathcal{K}} w_i(s') |Q(s', a') - Q'(s', a')|.$$

Since $\sum_i w_i(s') = 1$ and $w_i(s') \geq 0$, it holds that

$$\sum_{s',a'} P_\pi(s', a'|s, a) \sum_{i \in \mathcal{K}} w_i(s') = 1,$$

so the inequality simplifies to

$$\|\mathcal{T}Q - \mathcal{T}Q'\|_\infty \leq \gamma \|Q - Q'\|_\infty.$$

Therefore $\mathcal{T}$ is still a $\gamma$-contraction; moreover,

$$\mathcal{T}Q^\pi = T^\pi Q^\pi = Q^\pi,$$

because $\sum_i w_i(s') = 1$ for every $s'$. Hence iterative evaluation enjoys the standard error bound

$$\|Q_k - Q^\pi\|_\infty \leq \gamma^k \|Q_0 - Q^\pi\|_\infty.$$

**(D) Reduced interference and better optimization.** Let $g_t(\theta)$ be the gradient at step $t$ on parameters $\theta$. In a monolithic model trained on heterogeneous data, expected gradient interference $\mathbb{E}[\langle g_t^{(u)}, g_t^{(v)} \rangle]$ between disparate regimes $u \neq v$ can be negative, slowing learning. Routing with MoE sparsifies updates so that each expert's parameters receive gradients primarily from the subset of states they serve:

$$\theta_i \leftarrow \theta_i - \eta \, \mathbb{E}_{(s,a)} \big[ w_i(s) \, \nabla_{\theta_i} \ell_i(s, a) \big],$$

which lowers cross-regime interference and accelerates specialization. Empirically, this manifests as fewer dormant units and better plasticity; theoretically, it reduces the effective condition number of the local objectives, easing optimization.

**(E) Expressivity for continuous actions.** A single Gaussian policy is unimodal; many continuous-control tasks are multi-modal (e.g., equivalent symmetric actions, contact-rich choices). Even when the final policy is kept Gaussian for simplicity, routing over experts induces a piecewise-smooth mapping

$$\mu_{\text{final}}(s) \; = \; \sum_{i \in \mathcal{K}(s)} \tilde{w}_i(s) \, \mu_i(s), \qquad \Sigma_{\text{final}}(s) \; = \; \sum_{i \in \mathcal{K}(s)} \tilde{w}_i(s) \, \Sigma_i(s),$$

which can approximate mixtures or mode switches via state-dependent convex combinations, substantially more expressive than a single global head.

---

**Takeaway**

ScaleMoE improves performance by (i) matching local structure with specialized, lower-complexity experts (lower bias), (ii) averaging over multiple routed estimates (lower variance), (iii) preserving Bellman contraction under convex fusion (stability), and (iv) mitigating gradient interference through sparse, state-dependent updates (better optimization). Load-balancing and importance regularizers further prevent expert collapse, ensuring that increased capacity is effectively trained rather than left dormant.

---

# M. Broader Impact

Our proposed ScaleMoE method contributes to closing the gap between the scaling capabilities of deep RL and those of supervised learning domains. By enabling RL agents to effectively utilize larger models, we open the door to solving more complex control problems and multi-task scenarios that were previously out of reach due to capacity limitations. **Positive impacts** could include more capable robotic controllers that can handle diverse tasks (thanks to expert specialization) and improved sample efficiency through conditional reuse of knowledge. For example, a household robot could employ a ScaleMoE policy to skillfully handle a wide array of chores by activating different expert networks for cooking vs cleaning tasks, rather than requiring separate models for each. Additionally, our analysis of neuron utilization might inform better

neural architecture design beyond RL, highlighting the importance of activating as much of the network as possible (which could lead to generally more efficient deep learning models).

On the **downside**, larger models and MoE architectures do raise concerns about computational cost and energy usage. Training the ScaleMoE agents on complex benchmarks consumed considerable GPU hour, although it can be more efficient than naive scaling. Developers should weigh the improvements in performance against the environmental and financial costs of training such models.

Overall, we believe the benefits of more scalable and generalizable RL agents outweigh the potential downsides, and we advocate for continued research in efficient scaling techniques like MoE to push the frontiers of what RL can achieve.

