# OpenReview forum: "ScaleMoE: Mixture-of-Experts for Scalable Continuous Control in Actor-Critic Reinforcement Learning"
_ICML.cc/2026/Conference — ICML 2026 spotlight_

### Official Review · Reviewer_nFnC · 2026-02-16

**Soundness:** 4
**Presentation:** 4
**Significance:** 3
**Originality:** 3
**Overall Recommendation:** 5
**Confidence:** 4

**Summary:**

This paper presents a scalable mixture of experts model architecture which is applied to modern actor-critic methods for continuous control tasks. The paper explores using mixture of experts layers at both the output layer and feature layer, finding different optimal results for single and multi-task problems. The paper is thoroughly ablated, investigating the effects of different numbers of experts, different top-k experts, and the impact of auxiliary loss terms. This paper sets itself apart from prior work by applying itself to both the actor and critic networks. The paper is well-written, rigorous and makes an interesting contribution.

**Compliance With Llm Reviewing Policy:**

Affirmed.

**Final Justification:**

This was a very strong paper, which was very well-written, thoroughly evaluated, and justified, and provides a high-performing practical method. The rebuttal addressed many of my concerns; however, I do not feel the contributions in this paper warrant a score higher than my previous score of 5. Therefore, I will maintain my "accept" rating.

**Key Questions For Authors:**

**Q1 (Updates-To-Data Ratio) -** For single task, you mention that you altered the update-to-data ratio from the original value and justify this decision in Appendix I. On these results, however, do you report the original algorithm's results with the new update-to-data ratio, or did you test both and use the best one? If you change the original algorithm's hyperparameters, I think it is important that you show this is not the reason behind the performance increase.

**Q2 (Higher Dimensional Data) -** In your limitations, you mention your work may not be easily applicable to higher-dimensional inputs. Could you elaborate on this? What problems do you foresee if one tried to implement this naively? While this may not be necessary, I'd love to see this work compared to [1] on a benchmark like Atari.

**Q3 (Auxiliary Loss Sensitivity) -** How sensitive were the auxilary loss parameters? Did you test any other values?

[1] Obando-Ceron, Johan, et al. "Mixtures of experts unlock parameter scaling for deep rl." arXiv preprint arXiv:2402.08609 (2024).

**Limitations:**

yes

**Strengths And Weaknesses:**

**Strength 1 (Rigor) -** This paper is very rigorous, with numerous ablations discussing almost everything I see as relevant. The paper also provides high-quality plots, and details around training such as resources used and model size comparison.

**Strength 2 (Multiple Tasks and Algorithms) -** This paper explores both the single and multi-task settings, using different algorithms, adding to the reliability of the method.

**Strength 3 (Ease of understanding) -** I found the paper clear, concise, and easy to read and understand. The paper flows well.

**Weakness 1 (Novelty) -** While I think the idea is interesting, I think it is fairly similar to existing work on MoEs. I do believe this work has novelty, but not in a groundbreaking way.

**Weakness 2 (Single-Task Environments) -** Although I was impressed by the different algorithms applied to both the single and multi-task settings, I would've appreciated seeing more than 7 single-task environments. This is quite small in comparison to other similar work.

---

> ### Author Rebuttal · Authors · 2026-03-31
>
> **We sincerely thank Reviewer nFnC for the valuable comments!**
>
> ## Seeing more than 7 single-task environments
>
> We apologize for any confusion regarding the experimental scope. Actually, our evaluation encompasses **77 continuous control environments in total**, comprising **50 MetaWorld** tasks, **20 HumanoidBench** tasks, and **7 DeepMind Control Suite** tasks.
>
> The comprehensive benchmark covering MetaWorld and HumanoidBench, which includes diverse manipulation and locomotion tasks, constitutes a substantially larger and more rigorous experimental protocol in Fig.3 and Fig.4 in our paper, respectively. Detailed training curves on each env are also provided in Fig.32 to Fig.33 in our paper.
>
> ## Updates-To-Data Ratio
>
> We appreciate this important methodological question. **To ensure absolute fairness, both SimBa and ScaleMoE (SimBa-based) were trained with the identical modified update-to-data (UTD) ratio** reported in Appendix I. We did not cherry-pick hyperparameters by testing multiple UTD values and selecting the best performer for each method individually; instead, we standardized the UTD modification across all methods to isolate the architectural contribution of sparse routing.
>
> Notably, **increasing the UTD ratio benefits SimBa itself.** Indeed, the original SimBa paper demonstrates that higher UTD improves performance (Figure 14 in [1]), so our modification does not disadvantage the baseline. The rationale for deviating from the original default is that **ScaleMoE introduces substantial additional capacity** (e.g., 4–64 experts) compared to the monolithic baseline, necessitating more gradient steps to optimize the expanded parameter space and stabilize the gating networks. This ensures both methods reach their respective convergence plateaus rather than comparing an undertrained high-capacity model against a baseline that has already saturated at a lower UTD.
>
> This aligns with standard protocol in scaling studies: when increasing model capacity, one must proportionally increase the optimization budget (gradient steps) to reveal the architecture’s true potential rather than artificially handicapping the larger model with insufficient updates. Thus, the performance gains reported reflect genuine architectural advantages of conditional computation, not artifacts of asymmetric hyperparameter tuning.
>
> ## Higher Dimensional Data
>
> The limitation regarding high-dimensional inputs stems from our architectural choice of independent encoders per expert. For image inputs (e.g., Atari frames), each expert would require its own convolutional encoder, causing the total parameter count to scale linearly with the number of experts. We will explore using shared vision encoder in the future to address this problem.
>
> ## Compared to Obando’s softmoe on a benchmark like Atari
>
> We implement our method based on DQN and compare with Obando’s softmoe and DQN baseline on three pixel-based Atari tasks. We set the number of experts to 8 for both MoE algorithms. Results show comparable overall performance, with task-specific variations: performance is similar on Pong, ScaleMoE achieves better results on Qbert, while SoftMoE performs better on SpaceInvaders. This suggests that both methods effectively utilize expert ensembles in discrete action spaces. Training curves could be found in Fig.7 in our [anonymous link](https://anonymous.4open.science/r/ICML2026-Submision3437-1D6F/).
>
> ## Auxiliary Loss Sensitivity
>
> Below we show the final IQM performance with 95% confidence interval (in the bracket) across 8 seeds of different **regularization weight** $\lambda$  after training 500k env steps on MetaWorld. The ablation confirms stable performance across two orders of magnitude, indicating the parameter **requires minimal tuning effort**. Training curves could be found in Fig.4 in our [anonymous link](https://anonymous.4open.science/r/ICML2026-Submision3437-1D6F/).
>
> In our experiments, we use a single fixed $\lambda$=0.01  across all tasks (MetaWorld 50 + Humanoid 20 + DMC 7) without any per-environment tuning. In practice, one could start with a minimal value and increase only as needed to ensure relatively balanced expert activation(Fig.15 in our paper).
>
> | $\lambda$ | Final IQM |
> | --- | --- |
> | 0.005 | 0.86 [0.83, 0.89] |
> | 0.01 | 0.86 [0.84, 0.87] |
> | 0.05 | 0.85 [0.81, 0.87] |
> | 0.1 | 0.80 [0.78, 0.84] |
> | 1 | 0.74 [0.71, 0.78] |
>
> [1] SimBa: Simplicity Bias for Scaling Up Parameters in Deep Reinforcement Learning

---

> > ### Author Rebuttal · Reviewer_nFnC · 2026-04-02
> >
> > Thank you for the clarifications on the benchmarks, UTD ratio, hyperparameter sensitivity, and higher-dimensional input. Many of my concerns were addressed. The results with DQN were a little lacklustre - DQN is an extremely old algorithm, and better/faster alternatives exist [1]. Furthermore, when running Atari experiments in future, consider using the recommendations of [2]. That said, these experiments were extras, and I still appreciate them.
> >
> > [1] Clark, Tyler, et al. "Beyond The Rainbow: High Performance Deep Reinforcement Learning on a Desktop PC." Forty-second International Conference on Machine Learning.
> > [2] Aitchison, Matthew, Penny Sweetser, and Marcus Hutter. "Atari-5: Distilling the arcade learning environment down to five games." International Conference on Machine Learning. PMLR, 2023.

---

> > > ### Author Response · Authors · 2026-04-06
> > >
> > > We sincerely thank you for your thorough and constructive feedback! We are delighted to have addressed your concerns. We will explore the design and application of MoE in discrete-action setttings in the future on the more advanced algorithms as you suggested.

---

### Official Review · Reviewer_QNJk · 2026-03-02

**Soundness:** 2
**Presentation:** 3
**Significance:** 3
**Originality:** 2
**Overall Recommendation:** 5
**Confidence:** 4

**Summary:**

This paper proposes ScaleMoE, an architecture that integrates Mixture-of-Experts modules into both the actor and critic networks of continuous-control actor-critic RL agents. The authors present two gating variants: output-level gating, which aggregates per-expert policy parameters and Q-values, and feature-level gating, which mixes expert representations before a shared output head. Comprehensive experiments and analysis on various environments show the scaling capacity and stability of ScaleMoE.

**Compliance With Llm Reviewing Policy:**

Affirmed.

**Final Justification:**

The authors' rebuttal has fully addressed my concerns. The paper is well-writen and shows strong empirical insights, further validate the scalability and promising potential of MoE in the RL field. I believe the paper should be accepted.

**Key Questions For Authors:**

1. Could you provide monolithic baselines at intermediate widths or intermediate number of parameters? This would establish obvious comparison between ScaleMoE and SimBa or BRC.

2. What is the rationale for using shared gating scores for both the Actor and Critic in Output-Level Gating? Have you considered comparing this to decoupled gating networks, where the Actor is conditioned on $s$ while the Critic is conditioned on $(s,a)$?

3. Could you summarize the key differences between the proposed ScaleMoE and existing methods like [1], which also investigate the effectiveness of MoE for scaling deep RL?


[1] Obando-Ceron et al., Mixtures of Experts Unlock Parameter Scaling for Deep RL

**Limitations:**

yes

**Strengths And Weaknesses:**

Strengths:

1. The method serves as a drop-in module to different RL algorithms and is easy to follow.

2. The paper conducts comprehensive experiments and analysis to the promising of introducing MoE to scale deep RL.

3. The paper investigates dormant neuron ratios, expert activation patterns, gradient interference, critic target variance, and activation entropy, providing actionable insights into why MoE helps.


Weaknesses:

1. The proposed two variants are natural and straightforward, representing a simple integration of standard MoE implementations that lacks novelty.

2. The compared monolithic baselines are tested at only one matched width (1472 for SimBa and 4096 for BRC, Figure 2), making it unclear how the monolithic scaling curve compared to ScaleMoE’s.

---

> ### Author Rebuttal · Authors · 2026-03-31
>
> **We sincerely thank Reviewer QNJk for the valuable comments!**
>
> ## Monolithic baselines at intermediate widths
>
> We provide intermediate monolithic baselines (BRC with increasing widths: 512/1024/2048/4096) alongside ScaleMoE variants (4/8/16/32/64 experts at fixed width 512) to enable direct comparison at matched parameter budgets. The table below reveals starkly different scaling behaviors: **ScaleMoE exhibits consistent monotonic improvement, outperforming** naive width scaling. Specifically, ScaleMoE with only 16 experts already surpasses the standard BRC baseline (width 4096). This comparison establishes that ScaleMoE's gains arise from **efficient conditional computation via sparse routing** rather than merely increasing parameter count through feedforward width expansion, validating that MoE architectures scale more effectively than monolithic widening.  Training curves could be found in Fig.6 in our [anonymous link](https://anonymous.4open.science/r/ICML2026-Submision3437-1D6F/)
>
>
> |  | BRC-512 | BRC-1024 | BRC-2048 | BRC-4096 |
> | --- | --- | --- | --- | --- |
> | **Final IQM** | 0.82 [0.80, 0.85] | 0.84 [0.84, 0.85] | 0.85 [0.83, 0.86] | 0.88 [0.84, 0.90] |
>
> |  | ScaleMoE-4 | ScaleMoE-8 | ScaleMoE-16 | ScaleMoE-32 | ScaleMoE-64 |
> | --- | --- | --- | --- | --- | --- |
> | **Final IQM** | 0.86 [0.81, 0.88] | 0.86 [0.84, 0.87] | 0.90 [0.88, 0.93] | 0.91 [0.89, 0.94] | 0.92 [0.87, 0.96] |
>
>
> ## Gating network conditioning
>
> We apologize for the imprecision in the original manuscript; we have corrected this in the revision to clarify that the Actor and Critic use decoupled gating networks with distinct inputs. Specifically, the Actor's gating is conditioned solely on the state $\mathbf{s}$, while the Critic's gating is conditioned on the state-action pair $(\mathbf{s}, \mathbf{a})$, as the Q-function must route based on the specific action being evaluated to properly assess its value. This architectural choice aligns with standard actor-critic design and ensures appropriate expert selection for policy generation versus value estimation. We thank the reviewer for helping us clarify this architectural detail.
>
> ## The key differences between  ScaleMoE  and baseline
>
> | **Feature** | **ScaleMoE** | **Obando-Ceron et al. (2024)** |
> | --- | --- | --- |
> | **Action Space** | Continuous (Gaussian mixture for policy) | Discrete (Q-learning with softmax expert selection) |
> | **MoE Integration** | Actor and Critic (MoE applied to both); Independent encoder for each expert; No slots concept | Value Function (MoE applied only to the critic); Shared encoders between experts; Multiple slot for each expert |
> | **Routing Location** | Top-K gating for both actor and critic networks | Soft routing (all experts are weighted for value function) |
> | **Gaussian Representation** | Each expert outputs mean and standard deviation (Gaussian) | No Gaussian mixture; uses softmax over Q-values for expert output |
> | **Multi-task Setting** | Yes, integrated with multi-task RL (task-specific expert specialization) | Primarily single-task RL (though adaptable to multi-task with expert specialization) |

---

> > ### Author Rebuttal · Reviewer_QNJk · 2026-04-01
> >
> > The experiments in this paper sufficiently validate the scalability and promising potential of MoE in the RL field. The authors' rebuttal has addressed my concerns, and I will keep my score in support of accepting this paper.

---

> > > ### Author Response · Authors · 2026-04-06
> > >
> > > We sincerely thank you for your thorough and constructive feedback! We are delighted to have addressed your concerns, and we will revise our paper according to your suggestions.

---

### Official Review · Reviewer_t3U8 · 2026-03-10

**Soundness:** 4
**Presentation:** 4
**Significance:** 4
**Originality:** 3
**Overall Recommendation:** 6
**Confidence:** 4

**Summary:**

This paper proposes ScaleMoE, a Mixture-of-Experts (MoE) architecture to improve the scalability of actor–critic reinforcement learning algorithms. The key idea is to replace the traditional monolithic actor and critic networks with a set of expert networks combined through a learned gating mechanism. The authors explore two integration strategies (feature-level gating and output-level gating) to incorporate expert routing into policy and value function learning. The approach is instantiated on top of strong RL baselines (SimBa for single-task learning and BRC for multi-task learning) and evaluated on a range of benchmarks including DeepMind Control Suite, MetaWorld, and HumanoidBench. Experimental results show that increasing the number of experts consistently improves performance and can outperform larger monolithic networks with comparable or even larger parameter budgets.

**Compliance With Llm Reviewing Policy:**

Affirmed.

**Final Justification:**

The authors' rebuttal has addressed all my concerns. The paper presents a well-motivated approach to an important problem in RL scaling and is supported by strong, highly comprehensive empirical results.

**Key Questions For Authors:**

How sensitive is the method to the architecture of the gating network?

**Limitations:**

yes

**Strengths And Weaknesses:**

### Strengths

- Addresses an important and timely problem in RL scaling. The question of how to scale neural network capacity in RL remains largely unresolved compared to fields like NLP and CV. This paper tackles this issue directly and explores conditional computation as an alternative scaling axis instead of simply increasing network width or depth.

- Clear and well-motivated idea. The motivation behind using mixture-of-experts in actor–critic architectures is intuitive and well explained. The authors clearly discuss why monolithic networks may struggle to fully utilize additional parameters and how expert routing can potentially alleviate this limitation.

- Comprehensive evaluation setup. The paper evaluates the method on multiple widely used benchmarks and considers both single-task and multi-task scenarios, demonstrating the approach's versatility.

- Strong empirical performance. The experimental results consistently demonstrate improvements over strong baselines across multiple environments. Rather than comparing with generic actor–critic models, the authors build on top of SimBa and BRC, which are already designed to address scaling challenges in RL. This makes the improvements more meaningful and strengthens the empirical claims.

- Insightful analysis. The paper includes additional analysis on expert utilization and network plasticity. In particular, the authors examine expert routing patterns and dormant neuron ratios, which provide useful insights into how the mixture-of-experts architecture utilizes model capacity.

### Weaknesses

- The mixture-of-experts design introduces additional architectural components and hyperparameters, such as the gating configurations, which may require some tuning in practice.

- The experiments primarily focus on environments with state-based inputs. While the current experimental evaluation is already quite comprehensive, it would be interesting to explore more perception-heavy RL settings (e.g., pixel-based tasks) in future work.

---

> ### Author Rebuttal · Authors · 2026-03-31
>
> **We sincerely thank Reviewer t3U8 for the valuable comments!**
>
> ## The gating configurations may require some tuning
>
> We perform gating network size ablation and auxiliary regularization ablation.
>
> Below we show the final IQM performance with 95% confidence interval (in the bracket) across 8 seeds of ScaleMoE with different gating hidden dimension after training 500k env steps on MetaWorld.  It shows that performance remains **stable** across gating hidden dimensions, with IQM scores varying within $\pm 2\%$ and overlapping confidence intervals. This indicates that while the gating mechanism introduces additional capacity, its architectural configuration is **not a sensitive hyperparameter**. Training curves could be found in Fig.5 in our [anonymous link](https://anonymous.4open.science/r/ICML2026-Submision3437-1D6F/).
>
> | gating hidden dim | Final IQM |
> | --- | --- |
> | 64 | 0.86 [0.84, 0.87] |
> | 256 | 0.87 [0.85, 0.90] |
> | 1024 | 0.87 [0.86, 0.89] |
>
> Below we show the final IQM performance with 95% confidence interval (in the bracket) across 8 seeds of different **regularization weight** $\lambda$  after training 500k env steps on MetaWorld. The ablation confirms stable performance across two orders of magnitude, indicating the parameter **requires minimal tuning effort**. Training curves could be found in Fig.4 in our [anonymous link](https://anonymous.4open.science/r/ICML2026-Submision3437-1D6F/).
>
> In our experiments, we use a single fixed $\lambda$=0.01  across all tasks (MetaWorld 50 + Humanoid 20 + DMC 7) without any per-environment tuning. In practice, one could start with a minimal value and increase only as needed to ensure relatively balanced expert activation(Fig.15 in our paper).
>
> | $\lambda$ | Final IQM |
> | --- | --- |
> | 0.005 | 0.86 [0.83, 0.89] |
> | 0.01 | 0.86 [0.84, 0.87] |
> | 0.05 | 0.85 [0.81, 0.87] |
> | 0.1 | 0.80 [0.78, 0.84] |
> | 1 | 0.74 [0.71, 0.78] |
>
> ## Pixel-based tasks
>
> We implement our method based on DQN and compare with Obando’s softmoe and DQN baseline on three pixel-based Atari tasks. We set the number of experts to 8 for both MoE algorithms. Results show comparable overall performance, with task-specific variations: performance is similar on Pong, ScaleMoE achieves better results on Qbert, while SoftMoE performs better on SpaceInvaders. This suggests that both methods effectively utilize expert ensembles in discrete action spaces. Training curves could be found in Fig.7 in our [anonymous link](https://anonymous.4open.science/r/ICML2026-Submision3437-1D6F/).

---

> > ### Author Rebuttal · Reviewer_t3U8 · 2026-04-03
> >
> > I thank the authors for their thorough rebuttal. The additional ablations sufficiently address my concerns about hyperparameter sensitivity and demonstrate strong robustness. The added results on pixel-based Atari tasks strengthen the practical relevance and generality of the method. I have increased my score accordingly.

---

> > > ### Author Response · Authors · 2026-04-06
> > >
> > > We sincerely thank you for your thorough and constructive feedback! We are delighted to have addressed your concerns, and we will revise our paper according to your suggestions.

---

### Official Review · Reviewer_VVgn · 2026-03-10

**Soundness:** 3
**Presentation:** 3
**Significance:** 2
**Originality:** 2
**Overall Recommendation:** 5
**Confidence:** 4

**Summary:**

The authors consider the problem of critic parameter scaling in continuous action off-policy RL (e.g. SAC or TD3), where temporal difference learning does not allow for naive increase of the network size. In particular, the authors consider scaling the critic via the mixture of experts (MoE) architecture, which was previously shown to achieve promising results in a different RL setup (discrete action, vision-based Atari). The work proposes design choices around gating and fusion of the outputs, and tests the proposed implementation in single and multi-task continuous action RL.

**Compliance With Llm Reviewing Policy:**

Affirmed.

**Final Justification:**

The rebuttal and added experiments have largely resolved the concerns I raised in my original review. Thank you for addressing my questions.

While I do not view this work as having sufficient novelty or broad impact for a spotlight/oral recommendation, I believe it makes a solid contribution by investigating the effectiveness of MoE networks for scaling critics, and it connects well to prior work in this area. As such, I believe the paper should be accepted.

**Key Questions For Authors:**

My main questions are related to the relationship between this work and [1]:
1. Did the authors consider running a soft MoE module? Would it work as well as in [1]?
2. Did the authors test tokenization mechanisms introduced in [1]? They were shown to impact performance significantly when using MoE.
3. Maybe an ensemble of smaller Simba/BRO networks would be a good baseline to compare to?
4. The "auxiliary regularization" mechanisms are sensible, maybe the ablations on using those should be summarized in the main body as well? Additionally, it would be interesting to see the performance given different hyperparameters associated with these "auxiliary regularizations".

Nitpicks:

1. I think that the authors could sharpen the motivation in the introduction. Now it says that "monolithic scaling imposes a fundamental limit, causing performance to plateau as model size increases beyond a certain point". This is not a problem if the monolithic method solves the task. As such, I think that a more convincing framing for different architectures is compute or parameter efficiency.
2. Additionally, in the introduction the authors write: "naive width or parameter scaling of actor–critic networks can induce overfitting and instability, obscuring clear scaling laws. For example, (Andrychowicz et al., 2021; Bjorck et al., 2021) report that standard agents plateau or degrade as parameters increase". Andrychowicz work focuses on on-policy learning (e.g. PPO) so not sure if this is a perfect citation. Bjorck work is in off-policy and they are the first to show that normalization helps scaling in SAC-like algorithms. Finally, there are scaling law papers in off-policy RL [2,3].

[1] Obando-Ceron et al. "Mixtures of Experts Unlock Parameter Scaling for Deep RL". ICML 2024

[2] Rybkin et al. "Value-Based Deep RL Scales Predictably". ICML 2025

[3] Fu et al. "Compute-optimal scaling for value-based deep rl". NeurIPS 2025

**Limitations:**

yes

**Strengths And Weaknesses:**

Strengths:
1. Studying the most efficient ways of scaling the critic is timely and potentially impactful.
2. MoE was shown to work well in discrete action, vision-based RL so it is good to see somebody filling the knowledge gap.

Weaknesses:
1. Whereas I like the paper in general (I think that there should be a paper that studies MoE in the context of state-based continuous action RL), I think that the paper is a bit of a missed opportunity. Specifically, I think that the paper should do a better job in contextualizing the research outcomes with respect to prior work. For example, works like BRO/BRC or Simba/Simbav2 showed that the performance drop from naive scaling is associated with a variety of other factors, such as exploding gradient norms or value overestimation (and BRO/Simba architectures are shown to remedy that). Does the MoE address these issues as well? Would MoE work with vanilla MLPs (without BRO/Simba architectures)? I believe that answering these questions would enhance the overall quality of the manuscript and help contextualize the proposed method in relation to prior works. Having said this, I like the dormant neurons analysis; I just think that dormant neurons are less established as a problem in the context of continuous control.

---

> ### Author Rebuttal · Authors · 2026-03-31
>
> **We sincerely thank Reviewer VVgn for the valuable comments!**
>
> ## Does the MoE address the exploding gradient norms?
>
> Yes, ScaleMoE maintains gradient norm magnitudes comparable to BRC, confirming that sparse routing does not exacerbate gradient instability. See Fig.1 in our [anonymous link](https://anonymous.4open.science/r/ICML2026-Submision3437-1D6F/).
>
> ## Would MoE work with vanilla MLPs?
>
> Yes, in Fig.5 in our paper, we design an MLP-based ScaleMoE for experimental validation. The final results below show that while scaling a monolithic MLP network’s width to 4096 yields diminishing returns, our MLP-based ScaleMoE approach scaling the expert count to 32 achieves significantly better performance with notably fewer parameters, further justifying the efficiency and effectiveness of MoE-based scaling.
>
> | Model | Critic width | Experts | Final IQM |  |
> | --- | --- | --- | --- | --- |
> | MLP | 4096 | — | 0.82 [0.80, 0.84] |  |
> | MLP | 1024 | — | 0.83 [0.82, 0.85] |  |
> | MLP | 256 | — | 0.75 [0.71, 0.80] |  |
> | ScaleMoE (MLP) | 256 | 32 | 0.92 [0.90, 0.93] |  |
> | ScaleMoE (MLP) | 256 | 16 | 0.85 [0.83, 0.88] |  |
> | ScaleMoE (MLP) | 256 | 8 | 0.81 [0.77, 0.85] |  |
>
> ## Running a soft MoE module
>
> We implemented SoftMoE based on the BRC codebase.  Below we show the final IQM performance with 95% confidence interval (in the bracket) across 8 seeds of SoftMoE-BRC and our method ScaleMoE after training 500k env steps on MetaWorld. ScaleMoE consistently outperforms SoftMoE-BRC across all expert configurations. Currently the num_slots in SoftMoE is automatically determined as in the original paper [1] and could be very large under our large batch settings. Therefore, the unsatisfactory performance of it could derive from the unstable optimization. We will further investigate SoftMoE in continuous action spaces in our future work. Training curves could be found in Fig.2 in our anonymous link.
>
> | Num Expert | 4 | 8 | 16 | 32 |
> | --- | --- | --- | --- | --- |
> | SoftMoE BRC | 0.70 [0.68, 0.77] | 0.77 [0.75, 0.80] | 0.82 [0.81, 0.84] | 0.81 [0.76, 0.83] |
> | ScaleMoE (Our) | 0.86 [0.81, 0.88] | 0.86 [0.84, 0.87] | 0.90 [0.8, 0.93] | 0.91 [0.89, 0.95] |
>
> ## Tokenization mechanisms choice
>
> In [1], the SoftMoE formulation treats the token axis ($M$) as a generic routing index, not inherently tied to spatial dimensions. While vision applications naturally instantiate $M = h \times w$ from convolutional features, state-based RL operates on single observation vectors lacking intrinsic spatial factorization. Following standard practice in non-sequential settings, we assign the batch/parallel-env dimension to $M$, where each sample constitutes one $d$-dimensional token.  Alternative tokenizations (e.g., learned projections to pseudo-tokens) are valid design choices; our approach prioritizes compatibility with standard vector-valued RL pipelines while preserving the core dispatch/combine mechanism. We are happy to add a clarifying paragraph distinguishing between spatial tokenization for convolutional outputs and batch-indexed tokenization for vector states.
>
> ---
>
> ## Comparison with ensemble networks
>
> We implemented ensemble of BRC. Below we show the final IQM performance with 95% confidence interval (in the bracket) across 8 seeds of Ensemble-BRC and our method ScaleMoE after training 500k env steps on MetaWorld. ScaleMoE demonstrates favorable scaling properties that naive ensembling lacks. While Ensemble-BRC suffers from performance degradation as size increases, ScaleMoE achieves monotonic improvement. Training curves could be found in Fig.3 in our anonymous link.
>
> | Num Expert/Ensemble | 1 | 4 | 8 | 16 |
> | --- | --- | --- | --- | --- |
> | Ensemble BRC | 0.82 [0.80, 0.85] | 0.87 [0.77, 0.89] | 0.82 [0.77, 0.84] | 0.79 [0.75, 0.80] |
> | ScaleMoE (Our) | 0.82 [0.80, 0.85] | 0.86 [0.81, 0.88] | 0.86 [0.84, 0.87] | 0.90 [0.8, 0.93] |
>
> ## Auxiliary regularization ablation
>
> Below we show the final IQM performance with 95% confidence interval (in the bracket) across 8 seeds of different **regularization weight** $\lambda$  after training 500k env steps on MetaWorld. The ablation confirms stable performance across two orders of magnitude, indicating the parameter **requires minimal tuning effort**. Training curves could be found in Fig.4 in our anonymous link.
>
> In our experiments, we use a single fixed $\lambda$=0.01  across all tasks (MetaWorld 50 + Humanoid 20 + DMC 7) without any per-environment tuning. In practice, one could start with a minimal value and increase only as needed to ensure relatively balanced expert activation(Fig.15 in our paper).
>
> | $\lambda$ | Final IQM |
> | --- | --- |
> | 0.005 | 0.86 [0.83, 0.89] |
> | 0.01 | 0.86 [0.84, 0.87] |
> | 0.05 | 0.85 [0.81, 0.87] |
> | 0.1 | 0.80 [0.78, 0.84] |
> | 1 | 0.74 [0.71, 0.78] |
>
> ## Other advices
>
> We would sharpen the motivation and use the suggested more suitable citations to clarify our opinions.
>
> [1] Mixtures of Experts Unlock Parameter Scaling for Deep RL

---

> > ### Author Rebuttal · Reviewer_VVgn · 2026-04-03
> >
> > The rebuttal and added experiments have largely resolved the concerns I raised in my original review. Thank you for addressing my questions.
> >
> > While I do not view this work as having sufficient novelty or broad impact for a spotlight/oral recommendation, I believe it makes a solid contribution by investigating the effectiveness of MoE networks for scaling critics, and it connects well to prior work in this area. As such, I believe the paper should be accepted.

---

> > > ### Author Response · Authors · 2026-04-06
> > >
> > > We sincerely thank you for your thorough and constructive feedback! We are delighted to have addressed your concerns, and we will revise our paper according to your suggestions.

---

### Decision · Program_Chairs · 2026-04-30

**Decision:**

Accept (spotlight)

**Comment:**

This study introduces a scalable mixture-of-experts (MoE) architecture designed for modern actor-critic methods applied to continuous control tasks. The authors investigate the application of MoE at both the output layer and the feature layer, revealing that optimal configurations differ between single-task and multi-task scenarios. The experimental results presented provide strong evidence of MoE’s scalability and its promising potential within reinforcement learning. All reviewers recommend accepting this paper.